# Dysfunctional oxidative phosphorylation shunts branched-chain amino acid catabolism onto lipogenesis in skeletal muscle

Cristina Sánchez-González[1] (ID), Cristina Nuevo-Tapioles[1,2,3] (ID), Juan Cruz Herrero Martín[1], Marta P Pereira[1] (ID), Sandra Serrano Sanz[1], Ana Ramírez de Molina[4], José M Cuezva[1,2,3] (ID) & Laura Formentini[1,2,3,*] (ID)

## Abstract

It is controversial whether mitochondrial dysfunction in skeletal muscle is the cause or consequence of metabolic disorders. Herein, we demonstrate that *in vivo* inhibition of mitochondrial ATP synthase in muscle alters whole-body lipid homeostasis. Mice with restrained mitochondrial ATP synthase activity presented intrafiber lipid droplets, dysregulation of acyl-glycerides, and higher visceral adipose tissue deposits, poising these animals to insulin resistance. This mitochondrial energy crisis increases lactate production, prevents fatty acid β-oxidation, and forces the catabolism of branched-chain amino acids (BCAA) to provide acetyl-CoA for *de novo* lipid synthesis. In turn, muscle accumulation of acetyl-CoA leads to acetylation-dependent inhibition of mitochondrial respiratory complex II enhancing oxidative phosphorylation dysfunction which results in augmented ROS production. By screening 702 FDA-approved drugs, we identified edaravone as a potent mitochondrial antioxidant and enhancer. Edaravone administration restored ROS and lipid homeostasis in skeletal muscle and reinstated insulin sensitivity. Our results suggest that muscular mitochondrial perturbations are causative of metabolic disorders and that edaravone is a potential treatment for these diseases.

**Keywords** Acetyl-CoA; ATP synthase; edaravone; insulin resistance; mitochondria
**Subject Categories** Metabolism; Musculoskeletal System; Organelles
**The EMBO Journal (2020) 39: e103812**

## Introduction

Obesity is a complex chronic condition that affects all organ systems and increases the rate of premature mortality (Spiegelman & Flier, 2001; Stefan *et al*, 2017). Increased visceral white adipose tissue (v-WAT) is associated with elevated blood levels of nonesterified free fatty acid (FFA), which in turn may result in insulin resistance (IR) in peripheral insulin target tissues such as skeletal muscle (Skm) (Despres & Lemieux, 2006). Moreover, WAT and Skm are endocrine organs that release and respond to hormones, a function that contributes to chronic inflammation associated with metabolic diseases (Pedersen & Febbraio, 2012; Stanford *et al*, 2015; Ciaraldi *et al*, 2016).

Interest in a role for mitochondria in the setting of metabolic disorders has increased in response to growing evidence linking mitochondrial dysfunction with pathology (Friedman & Nunnari, 2014; Shadel & Horvath, 2015; Kauppila *et al*, 2017). Mitochondria are essential in maintaining cell homeostasis by controlling bioenergetics, immunity, intracellular signaling, and cell death (Spinelli & Haigis, 2018). Furthermore, these organelles are involved in coordinating cellular adaptation to stressors and nutrient availability, and regulating glucose, amino acid, and lipid metabolism (Liesa & Shirihai, 2013; Vyas *et al*, 2016; Garcia-Bermudez *et al*, 2018). Depending on the disposal of ATP, TCA intermediates, and reduced coenzymes (NADH, FADH$_2$), different intramitochondrial dehydrogenases transfer electrons to the respiratory complexes of the electron transport chain (ETC) to generate the proton electrochemical gradient used for ATP synthesis in oxidative phosphorylation (OXPHOS). Since Skm is the largest oxidative and insulin-sensitive organ in mammals, understanding this mitochondrial-mediated metabolic flexibility may reveal new therapeutic strategies for diseases characterized by whole-body dysregulation of glucose and lipid metabolism.

Structural and functional perturbations in Skm mitochondria have been associated with the onset and complications of metabolic diseases (Kelley *et al*, 2002; Lowell & Shulman, 2005; Sivitz & Yorek, 2010; Hesselink *et al*, 2016). Reduced OXPHOS gene and protein expression has been described in response to genetic and nutritional obesity (Mootha *et al*, 2003; Patti *et al*, 2003; Sparks *et al*, 2005; Wang *et al*, 2010). In fact, a reduction of up to 40% in

---

1 Departamento de Biología Molecular, Centro de Biología Molecular "Severo Ochoa" (CBMSO), Universidad Autónoma de Madrid, Madrid, Spain
2 Centro de Investigación Biomédica en red de Enfermedades Raras (CIBERER), ISCIII, Madrid, Spain
3 Instituto de Investigación Hospital 12 de Octubre, i+12, Universidad Autónoma de Madrid, Madrid, Spain
4 Molecular Oncology and Nutritional Genomics of Cancer Group, Instituto Madrileño de Estudios Avanzados (IMDEA) Food Institute, CEI UAM+CSIC, Madrid, Spain
*Corresponding author. Tel: +34 9119 64648; E-mail: lformentini@cbm.csic.es

the expression and activities of Skm respiratory complexes has been reported in T2D subjects (Kelley *et al*, 2002; Ritov *et al*, 2005; Formentini *et al*, 2017a). Skm from the same T2D individuals presented diminished-size fissioned mitochondria, particularly in subsarcolemmal fractions (Ritov *et al*, 2005), indicating perturbations in mitochondrial dynamics. Additionally, hereditable Skm mitochondrial dysfunctions have been identified by NMR spectroscopy *in vivo*, unveiling OXPHOS and TCA defects in offspring of T2D subjects (Petersen *et al*, 2005; Befroy *et al*, 2007).

A key unanswered question is whether the mitochondrial alterations observed in Skm from T2D subjects are secondary features of the dyslipidaemic environment or participate in the setting of dyslipidaemia. In fact, the exact mechanism linking mitochondrial activity with obesity and T2D remains to be clarified.

A pivotal regulator of mitochondrial function is the ATP synthase, an inner membrane (IMM) enzyme at the crossroads of bioenergetics, apoptosis, redox metabolism, and the shape of cristae (Sanchez-Arago *et al*, 2013; Spinelli & Haigis, 2018). In this study, we demonstrate that its activity plays a key role as a transducer in lipid metabolism. The *in vivo* inhibition of Skm ATP synthase triggers lipogenic reprogramming to an increased lipid synthesis in both muscle and WAT, causing these animals to develop T2D faster upon feeding them a high-fat diet (HFD). *In vivo* treatment with the mitochondrial enhancer edaravone restored lipid and glucose homeostasis in mice. Hence, we propose that mitochondrial activity is a key regulator of skeletal muscle metabolism and endocrine signaling.

# Results

### A mouse model for the *in vivo* impairment of Skm OXPHOS

In order to assess the role of OXPHOS on the pathophysiology of Skm lipid metabolism, we generated an inducible and tissue-specific mouse model that expressed the active form (Boreikaite *et al*, 2019) of the human ATP synthase inhibitor $ATPIF1_{H49K}$ (Formentini *et al*, 2014) in striatal muscle. Double transgenic Tet-On mice $ATPIF1_{H49K}|^{T/H}$ (Fig 1A) express $ATPIF1_{H49K}$ in ACTA-1-positive myocytes (Fig 1B–D), while no expression was observed in the brain, liver, or WAT (Fig 1D). Interestingly, the endogenous expression of mouse-ATPIF1 in Skm was not detectable (Fig 1D). $O_2$ consumption rate (OCR) in isolated mitochondria from Skm confirmed that state 3 (ADP-stimulated) but not uncoupled (FCCP-induced) respiration was significantly inhibited in $ATPIF1_{H49K}|^{T/H}$ mice when compared to that in control mice (Fig 1E), suggesting that the $ATPIF1_{H49K}$ effect was specific to ATP synthase. However, the reduction in total $O_2$ consumption (Fig 1E, left panel) indicated a lower mitochondrial respiration in $ATPIF1_{H49K}|^{T/H}$ mice (referred to from now on as $Low_{OXPHOS}$ mice).

We next sought to unveil the impact of limiting OXPHOS on the Skm proteome by performing iTRAQ quantitative proteomics on hindlimb muscles from wt and $Low_{OXPHOS}$ mice (Figs 1F–K, and EV1 and EV2). Of the 1,250 proteins identified, 2.1% were significantly downregulated and 7.7% were upregulated in $Low_{OXPHOS}$ fibers compared to their expression in fibers from wt animals (Fig 1F–H). Gene Set Enrichment (GSEA) bioinformatic analysis of the results revealed that 14 metabolic pathways were perturbed when OXPHOS was inhibited (Figs 1I and J, and EV1 and EV2). The

related enrichment scores (ES and NES) were particularly significant for the redox system, lipid metabolism, BCAA catabolism, and FAD-binding proteins (Fig 1J). Remarkably, the Skm iTRAQ $Low_{OXPHOS}$/wt ratio for most of the lipid oxidation and BCAA catabolism enzymes was significantly upregulated (Figs 1K, and EV1 and EV2).

### Skm-restrained OXPHOS alters lipid contents and metabolism

In line with the possible alterations in lipid metabolism, the $Low_{OXPHOS}$ mouse body weight was higher than in wt mice (Figs 2A and EV3A). Moreover, the hindlimb muscles of the transgenic mice, particularly the soleus oxidative muscle, were significantly whiter than the control (Fig 2B) due to a higher intramuscular infiltration of adipocytes (Fig 2C and D). Accordingly, the iTRAQ ratio (Fig 2E) and WB expression (Fig 2F) of Skm proteins related to lipid synthesis were higher in $Low_{OXPHOS}$ mice compared to wt littermates. In particular, expression in muscle of the lipogenic enzymes ATP citrate lyase (ACLY) and fatty acid synthase (FASN) was 130 and 210%, respectively, upregulated in mice with restrained OXPHOS (Fig 2F), suggesting altered *de novo* lipogenesis. Intriguingly, in $Low_{OXPHOS}$ mice ACLY resulted highly acetylated, what has been related to the stabilization and activation of the protein, promoting lipid biosynthesis [(Lin *et al*, 2013), Fig 2G]. Consistently, we found that acetyl-CoA accumulated in Skm of $Low_{OXPHOS}$ mice (Fig 2H). In Skm mitochondria, acetyl-CoA is preferentially generated as the end product of glucose metabolism, free fatty acid (FFA) β-oxidation, or BCAA catabolism (Pietrocola *et al*, 2015; Fig 2I). In order to unveil the primary source for acetyl-CoA accumulation in $Low_{OXPHOS}$ mice, we first investigated glucose metabolism. As previously reported (Formentini *et al*, 2012), the inhibition of mitochondrial ATP production (Fig 1E) caused a rewiring of energy metabolism through an increased aerobic glycolysis (Figs 2J and EV3B), with the aim to maintain Skm ATP levels (Fig EV3C). However, despite a slight increase in glucose uptake (Fig 2K), $ATPIF1_{H49K}$ expressing myocytes showed a reduced total oxidation of 14C(u)-glucose to $CO_2$ in comparison to controls (Fig 2L). This could be due to the role of acetyl-CoA as a metabolic sensor able to allosterically inactivate enzymes involved in its synthesis, such as the pyruvate dehydrogenase complex (PDH) (Pietrocola *et al*, 2015). Accordingly, we found that PDH was phosphorylated in $Low_{OXPHOS}$ mice (Fig 2M), and pyruvate was rerouted to lactate production (Fig 2N), thus ruling out glycolysis as the main source of acetyl-CoA during OXPHOS inhibition.

The *de novo* lipid synthesis intermediate malonyl-CoA is known to limit FFA degradation (Foster, 2012; Fig 2I). In line with this and with a previous report in human myotubes (Formentini *et al*, 2017a), FFA β-oxidation resulted in a 35% inhibition in myocytes derived from $Low_{OXPHOS}$ compared to those in wt mice (Fig 2O). This thus suggests that enhanced Skm BCAA utilization or other minor pathway fluxes [such as reductive glutaminolysis (Liu *et al*, 2016)] are responsible for the observed acetyl-CoA accumulation.

As a result of increased lipid synthesis along with a decrease in their catabolism, FFAs accumulated in muscle (Fig 2P). Because this dysregulation was accompanied by an increase in Skm glycerol levels (Fig EV3D), we thus reasoned that these events might contribute to altered intramuscular neutral lipid storages. Accordingly, optic (Fig 2Q and R) and electron microscopy (Fig 2S) images showed the presence of lipid droplets (LD; Fig 2Q–S) inside

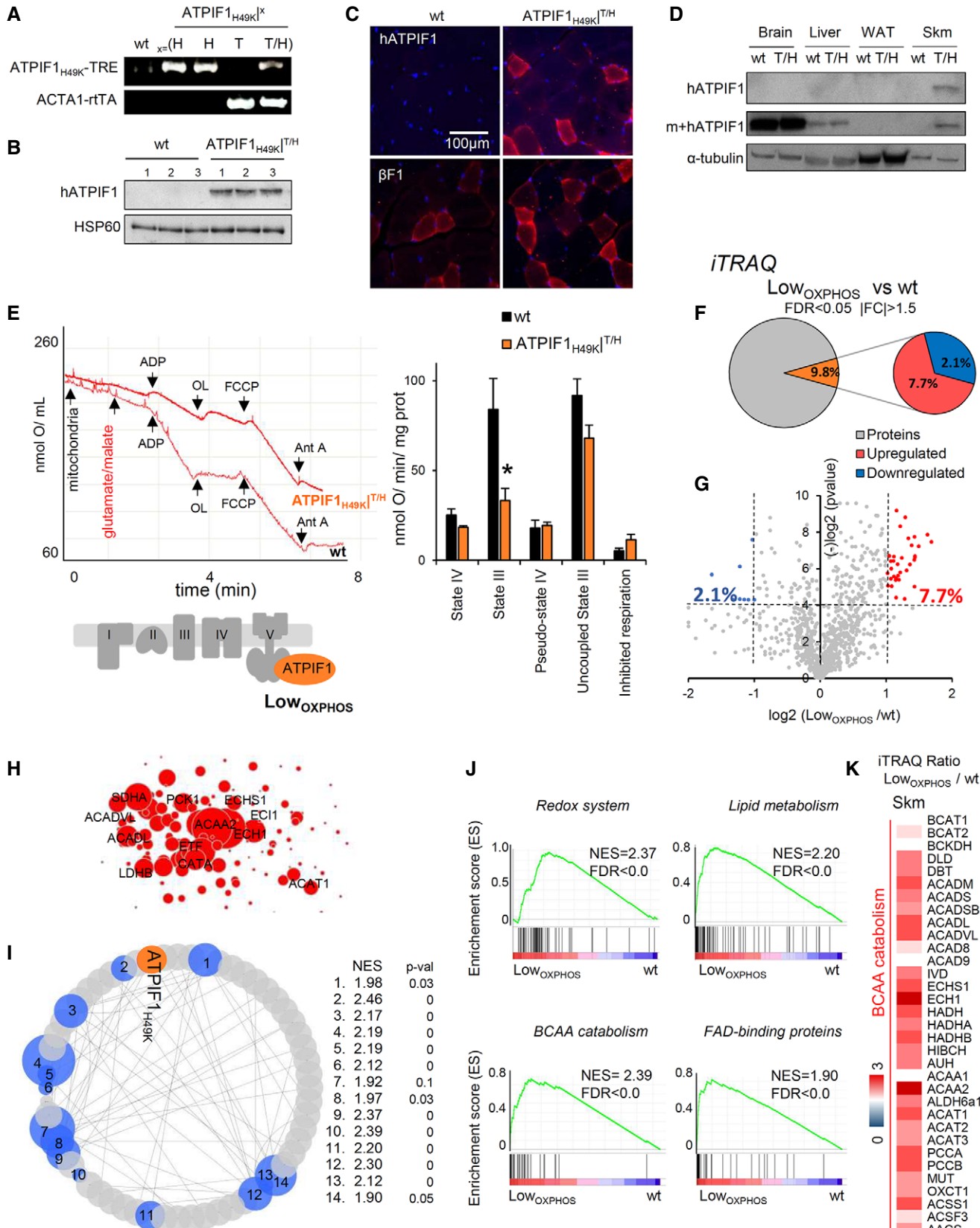

**Figure 1.**

**Figure 1.   A mouse model for the *in vivo* inhibition of Skm OXPHOS.**

A       PCR analysis of the human H49K variant of the ATPIF1 and rtTA constructs in wild-type (wt), ACTA1-rtTA (T), ATPIF1$_{H49K}$-TRE (H) or double transgenic (T/H) mice.
B–D    (B, D) WB expression of the human (h) or human + mouse (m + h) ATPIF1 protein in Skm (B, D), brain, liver, and WAT (D) extracts. hATPIF1 is only expressed in Skm from ATPIF1$_{H49K}$|$^{T/H}$ mice. HSP60 and α-tubulin are shown as loading controls. *n* = 3 mice/genotype. (C) Immunofluorescent staining of wt and ATPIF1$_{H49K}$|$^{T/H}$ hindlimb muscle slices with hATPIF1 and βF1 antibodies. Images are representative of 3 mice/genotype, 5 fields/mouse. hATPIF1 $_{H49K}$ is only expressed in T/H mice.
E       Polarographic profiles of isolated mitochondria from wt (lower trace) and ATPIF1$_{H49K}$|$^{T/H}$ (upper trace) animals. Histograms show a reduction in state 3 respiration consistent with the inhibition of ATP synthase in ATPIF1$_{H49K}$|$^{T/H}$ mice. Bars are the mean ± SEM of *n* = 3 mice/genotype, 3 traces/mouse; OL, oligomycin; Ant A, antimycin A.
F, G    Quantitative proteomic analysis (iTRAQ) of hindlimb Skm from wt and Low$_{OXPHOS}$ mice. The pie charts (F) and volcano plot (G) present the upregulated (log$_2$[Low$_{OXPHOS}$/wt] ≥ 1, red) or downregulated (log$_2$[Low$_{OXPHOS}$/wt] ≤ −1, blue) proteins. A (−)log *P*-value of > 4 was considered statistically significant.
H       Cytoscape representation of the most upregulated proteins in Skm from Low$_{OXPHOS}$ mice.
I       Cytoscape representation of GSEA analysis. Fourteen significantly altered pathways and their normalized enrichment score (NES) are shown. 1. Hallmarks of adipogenesis; 2. Go_Mitocondrial signal; 3. Kegg_TCA Cycle 4; Pyruvate Metabolism Reactome; 5. Hallmarks of OXPHOS; 6. Go_ETC; 7. Go_NADPH metabolism; 8. Kegg_Propionate metabolism; 9. Redox system; 10. Kegg_BCAA catabolism; 11. Go_Lipid metabolism; 12. Kegg_Lipid oxidation; 13. NEFA, Tg and ketone bodies; 14. Go_FAD-binding proteins. See also Fig EV1.
J       Enrichment score (ES) graphs from GSEA analysis.
K       Skm iTRAQ ratio of proteins from BCAA catabolism. Higher intensities of red or blue colors represent higher or lower Low$_{OXPHOS}$/wt expression ratios, respectively.

Data information: (F–K) Data are the mean ± SEM of *n* = 12 animals/genotype measured in 2 different iTRAQ analysis *P* < 0.05 when compared to wt by ANOVA and Student's *t*-test. See also Figs EV1 and EV2.
Source data are available online for this figure.

the oxidative polygonal myofibers of the soleus (Fig 2R) from Low$_{OXPHOS}$ but not wt mice.

To confirm the direct relationship between ATPIF1$_{H49K}$ and LD formation, we overexpressed the human protein in mouse C$_2$C$_{12}$ myocytes (Fig 2T). The resulting inhibition of the ATP synthase activity during starvation and palmitate supplementation triggered the development of a higher number of BODIPY-positive LDs in comparison to the control (Figs 2T and EV3E). Similar results were obtained by the pharmacological inhibition of the ATP synthase (5 μM oligomycin, Figs 2U and EV3F), indicating that this is a general trait of inhibiting CV. Interestingly, we did not find changes in the expression of proteins from mTOR or autophagy pathways in Low$_{OXPHOS}$ mice (Fig EV3G and H), suggesting that these processes are not involved. However, further studies would be required to exclude their participation in the observed lipogenesis.

In order to verify if lipid synthesis and accumulation upon OXPHOS inhibition may be related to augmented BCAA catabolism, we performed the same experiment in BCAA-free media (containing other AAs and palmitate as biosynthetic substrates). Upon this condition, no differences in LD formation were observed in the presence or absence of the ATP synthase inhibitor (Fig 2T). In line with this hypothesis, plasma levels of BCAA were augmented in Low$_{OXPHOS}$ mice compared to wt (Fig 3A), and the uptake of 14C(u)-leucine or 14C(u)-isoleucine increased in myocytes expressing ATPIF1$_{H49K}$ (Fig 3B). Interestingly, no differences in the Skm BCAA amounts were detected between the two genotypes (Fig 3C), suggesting an increase in BCAA muscular catabolism in Low$_{OXPHOS}$ mice. Consistently, in this situation, 14C(u)-BCAA oxidation to CO$_2$ was increased compared to control (Fig 3D). Moreover, when myocytes were administered with 14C(u)-leucine or 14C(u)-isoleucine and the lipid fraction extracted, a higher concentration of C$^{14}$-lipids was observed upon OXPHOS inhibition (Fig 3E), what suggests a higher BCCA catabolism and incorporation into lipids in Low$_{OXPHOS}$ mice.

## Mitochondrial-driven rewiring of Skm lipid and BCAA metabolism signals a hyperlipidemic phenotype

Despite the observed ATP synthase-dependent lipid dysregulation in muscle, the transgenic mice did not present alterations in blood glucose levels (Fig EV3I) or in tissue insulin or glucose sensitivity (Fig EV3J). Therefore, we next investigated the possibility that feeding animals a HFD may potentiate the phenotype in Low$_{OXPHOS}$ mice. After 60 days of HFD, the structure of Skm myofibers was altered, and intramuscular adipocyte storage became elevated in both wt and ATPIF1$_{H49K}$-expressing mice (Fig 3F). However, animals with restrained OXPHOS displayed a much stronger phenotype, with reduced soleus mass, shrinkage of fibers (Fig 3F, arrows), and greater adipocyte infiltration (Fig 3F, left panels) compared to those of wt animals. Therefore, we next sought to disclose the impact of muscle mass perturbations on the motor function of Low$_{OXPHOS}$ mice. No significant alterations in rotarod (Fig 3G) and open field (Figs 3H and EV3K) tests were noticed between wt and Low$_{OXPHOS}$ animals; however, upon ATP synthase inhibition mice displayed reduced performances in the grip force test (Fig 3I).

Concomitantly to these alterations in motor patterns, Low$_{OXPHOS}$ mouse weight was higher than that in control mice (Fig EV4A). Despite no significant changes in mouse food intake were observed between wt and Low$_{OXPHOS}$ mice (Fig EV4B), lipidomic analysis revealed that at day 60 of HFD, the Skm lipidomic profile was deeply altered (Fig 3J) as a consequence of ATP synthase inhibition. In particular, the Skm levels of ceramides and DAGs, lipid species related to the onset of IR (Szendroedi *et al*, 2014; Turpin-Nolan *et al*, 2019), were increased 130% in Low$_{OXPHOS}$ mice in comparison to levels in wt mice (Fig 3K). Consistent with our hypothesis, the observed OXPHOS-dependent alteration in Skm lipid profile led to elevated blood triglycerides (Fig 3L), and Skm showed reduced expression of GLUT1 and PPARβ/δ (Fig 3M). Moreover, and consistent with previous data in human myocytes derived from obese and T2D subjects (Formentini *et al*, 2017a), Skm TNFα levels were 270% upregulated and the pattern of myokines and cytokines altered when ATP synthase was inhibited (Fig 3N), what may contribute to inflammation associated with the IR setting (Ciaraldi *et al*, 2016). All these metabolic alterations led to the transgenic animals being prone to developing metabolic diseases (Fig 3O and P). Indeed, wt mice developed T2D at day 80–90 of HFD (Fig 3O), while the blood glucose levels (Fig 3O) and the insulin (ITT) and glucose (GTT) tolerance tests (Fig 3P) in Low$_{OXPHOS}$ mice indicated

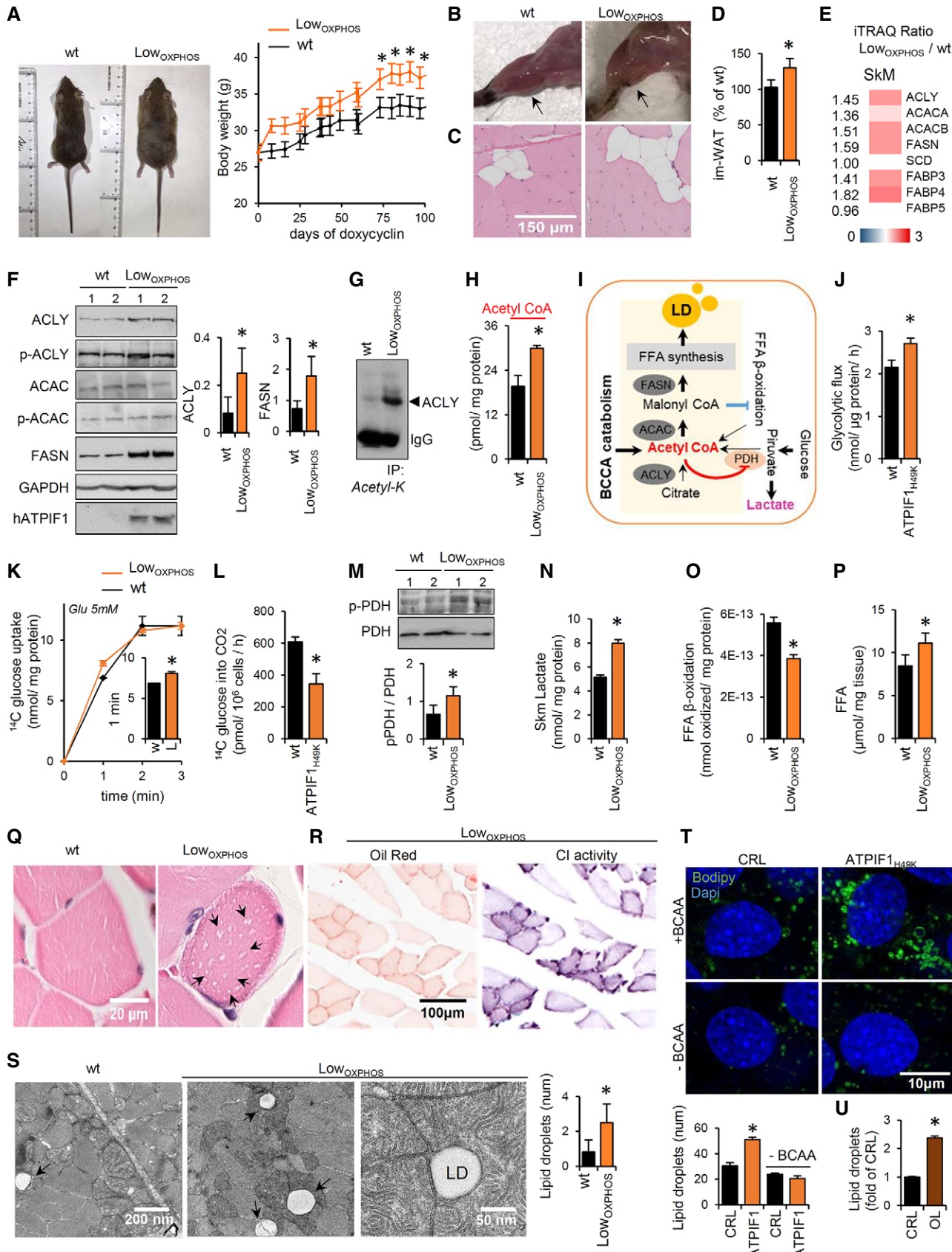

Figure 2.

**Figure 2. Perturbations in Skm OXPHOS alter lipid metabolism and storage.**

Data for wild-type (wt, black bars and traces) and Low$_{OXPHOS}$ (orange bars and traces) mice are shown.

A Representative images of mice and body weight graph following the expression of mitochondrial ATPIF1$_{H49K}$ (days of doxycycline) (wt, $n$ = 12; Low$_{OXPHOS}$, $n$ = 12).

B Representative images of hindlimb muscles. Arrows point to the soleus.

C, D Transversal slices of red fibers from soleus stained with hematoxylin/eosin (C). Higher im-WAT infiltrations in Low$_{OXPHOS}$ mice are shown. Quantifications in (D) (wt, $n$ = 6; Low$_{OXPHOS}$, $n$ = 6; 10 fields/mouse).

E, F iTRAQ ratio (E) and representative WB expression (F) of Skm proteins from *de novo* lipid synthesis. The expression of ATP citrate lyase (ACLY), acetyl-CoA carboxylase (ACAC), fatty acid synthase (FASN) and their phosphorylation (p) are shown. Two samples *per condition*; each sample contains protein extracts from 3 mice. GAPDH is shown as a loading control. Quantifications in lateral histograms (wt, $n$ = 6; Low$_{OXPHOS}$, $n$ = 6). In (E), a higher intensity of red color represents a higher Low$_{OXPHOS}$/wt expression ratio.

G Immunocapture (IP) of Skm acetylated proteins blotted with anti-ACLY antibody.

H Skm acetyl-CoA amounts (wt, $n$ = 8; Low$_{OXPHOS}$, $n$ = 8).

I Schematic representation of *de novo* FFA synthesis and the increase in LD. Elevated acetyl-CoA levels may be derived from dysregulation of the FFA β-oxidation, glycolysis, or BCAA metabolism.

J Myocyte rates of aerobic glycolysis to lactate production. Bars are the mean $\pm$ SEM of $n$ = 3 experiments, 9 replicas/condition.

K, L 14C(u)-glucose uptake (K) and oxidation to $CO_2$ (L) in myocytes expressing or not the ATP synthase inhibitor ATPIF1$_{H49K}$. Bars are the mean $\pm$ SEM of $n$ = 3 experiments, 6 replicas/condition.

M Representative WB expression of Skm PDH and its phosphorylation. Two samples *per condition*; each sample contains extracts from 3 mice. Quantification in histograms (wt, $n$ = 6; Low$_{OXPHOS}$, $n$ = 6).

N Skm levels of lactate (wt, $n$ = 6; Low$_{OXPHOS}$, $n$ = 6).

O FFA β-oxidation in primary cultures of myocytes derived from wt or Low$_{OXPHOS}$ mouse hindlimbs. Bars are the mean $\pm$ SEM of $n$ = 3 experiments, 9 replicas/condition.

P FFA amounts in Skm extracts (wt, $n$ = 8; Low$_{OXPHOS}$, $n$ = 8).

Q, R Transversal slices of soleus stained with hematoxylin/eosin (Q). Arrows indicate LDs in Low$_{OXPHOS}$ mice. Oil Red O staining and enzymatic activity of respiratory CI in transverse contiguous slices (R). Images are representative of $n$ = 4 mice/genotype, 10 images/mouse. Note the colocalization of oxidative fibers with LDs (R).

S Electron microscopy of transversal slices of the soleus. Intrafiber LDs surrounded by mitochondria in Low$_{OXPHOS}$ mice are shown. Histogram shows the quantifications of $n$ = 4 animals/genotype, 10 images/animal.

T LD formation upon ATP synthase inhibition (ATPIF1$_{H49K}$) in starved myocytes after 24 h of palmitate supplementation in the presence or absence of BCAA. Blue: DAPI, nuclei; green: BODIPY-positive LDs. Histograms show the quantification expressed as the number of LDs/nuclei. Bars are the mean $\pm$ SEM of $n$ = 3 experiments, 12 fields/condition. See Fig EV3E.

U LD formation upon ATP synthase inhibition (5 μM oligomycin) in myocytes after 24 h of palmitate supplementation. Histograms show the quantification expressed as fold of control of the BODIPY/DAPI fluorescence intensity. Seven fields/condition. See Fig EV3F.

Data information: Bars are the mean $\pm$ SEM of the indicated ($n$) mice/genotype *$P$ < 0.05 when compared to wt by ANOVA and Student's $t$-test. See also Fig EV3.

Source data are available online for this figure.

that T2D already occurred at day 60–70. Altogether, these experiments show that Skm OXPHOS activity plays a key role in the maintenance of whole-body glucose homeostasis.

In order to better understand the metabolic mechanisms leading to the observed phenotype, we looked for differentially regulated metabolic enzymes in our proteomic datasets. Interestingly, we found that whereas the BCAA catabolism proteins were significantly upregulated in muscle (Figs 1K, and EV1 and EV2), the WAT iTRAQ Low$_{OXPHOS}$/wt ratio showed the opposite trend for these same enzymes (Fig 4A). The increased Skm and inhibited WAT catabolism of BCAAs is a recently described marker of IR (Neinast *et al*, 2019). Consistently, odd fatty acids, which are substrates of the same pathway, are reduced in Skm and accumulated in WAT from restrained OXPHOS mice (Fig 4B). To note that Skm-specific impairment of OXPHOS also altered *de novo* lipid synthesis (Fig 4C) in WAT and modified the expression of protein from FA availability, lipid transport, and metabolism (Fig 4D). This may cause the weight of v-WAT to be higher in Low$_{OXPHOS}$ than in wt mice (Figs 4E and EV4C). These results suggest possible whole-body metabolic alterations or a mitochondrial-dependent cross-talk between muscle and adipose tissue (Pedersen & Febbraio, 2012) that deserve further investigation. Quantitative lipidomic analysis (Fig 4F) confirmed a significant upregulation of total TAGs (Figs 4G and EV4D) and DAGs (Figs 4G and EV4E) and saturated DAGs (Fig EV4F) in v-WAT from Low$_{OXPHOS}$ mice compared to that in wt mice.

## The lipogenic switch alters the redox system and lipid-related OXPHOS components

The observed increase in saturated lipids in WAT (Fig EV4F) and Skm (Figs 4H and EV4G) from mice with restrained OXPHOS raised the possibility that augmented ROS production in these mice may mediate these effects (Ayala *et al*, 2014; Shadel & Horvath, 2015). Inhibition of ATP synthase triggers the accumulation of ROS in cancer (Formentini *et al*, 2012). Indeed, in our model, mitochondrial ROS were 40% higher in myocytes expressing ATPIF1$_{H49K}$ than levels in the control (Fig 4I). Moreover, the production of 4-hydroxynonenal (4HN), a marker of ROS-dependent lipid peroxidation (Ayala *et al*, 2014), was augmented in Skm from Low$_{OXPHOS}$ mice compared to that in wt mice (Fig 4J). Consistent with the role of NADPH as a cofactor involved in detoxification of lipid ROS (Stockwell *et al*, 2017) and as a result of the ROS cascade, we observed a general alteration in the Redox system (Figs 4J and EV1) and NADP/NADPH ratio (Fig EV5A) in these animals.

In order to unveil the mitochondrial site(s) of origin for ROS production, we studied the direct effect of ATPIF1$_{H49K}$ on the OXPHOS proteins and related systems. Consistent with the $O_2$ consumption data (Fig 1E), no changes were observed in the Skm expression of different subunits from the respiratory CI, CIII, CIV, and CV between Low$_{OXPHOS}$ and wt genotypes (Fig 4K). However, SDHA and SDHB, belonging to CII, and subunits from the electron transfer flavoprotein ETF were significantly overexpressed upon

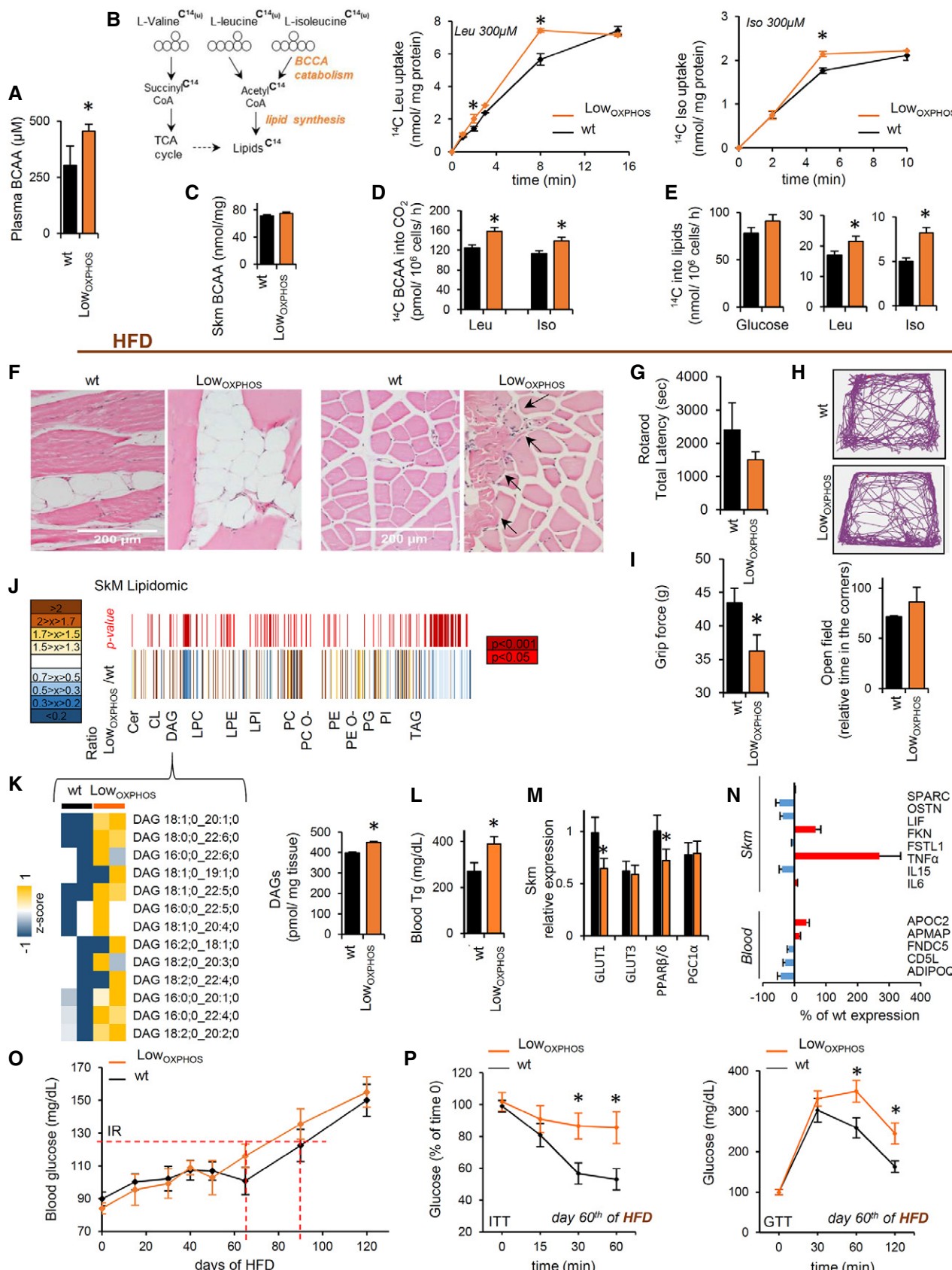

**Figure 3.**

◄

**Figure 3. OXPHOS-mediated alterations in lipid and amino acid metabolism contribute to IR.**

Data on wild-type (wt, black bars and traces) and Low$_{OXPHOS}$ (orange bars and traces) mice are shown.

A–E (A, C) Plasma (A) and Skm (C) BCAA levels in wt and Low$_{OXPHOS}$ mice (wt, $n = 4$; Low$_{OXPHOS}$, $n = 4$). (B, D, E). Scheme of the BCAA catabolism pathway (B, left panel) and 14C(u)-leucine and 14C(u)-isoleucine uptake (B, right panels), oxidation (D) and incorporation into lipids (E) in myocytes expressing or not ATPIF1$_{H49K}$. Bars are the mean ± SEM of $n = 3$ experiments, 6 replicas/condition.

F Longitudinal (left panels) and transversal (right panels) slices of soleus stained with hematoxylin/eosin. im-WAT infiltrations and fiber shrinkage (arrows) in HFD-fed Low$_{OXPHOS}$ mice (wt + HFD, $n = 4$; Low$_{OXPHOS}$ + HFD, $n = 4$; 10 fields/mouse).

G–I Motor function tests. Rotarod (G; wt, $n = 7$; Low$_{OXPHOS}$, $n = 6$), open field (H; wt, $n = 4$; Low$_{OXPHOS}$, $n = 4$) and grip force (after fatigue, I; wt, $n = 7$; Low$_{OXPHOS}$, $n = 6$) tests. In (H), lines represent the movement of the mice in the cage. The histogram represents the relative time that wt and Low$_{OXPHOS}$ mice expended in the corners.

J, K Quantitative Skm lipidomics at day 80 of HFD (J). The color scale (brown to blue) in the heat map represents Low$_{OXPHOS}$/wt amounts of detailed lipid species. A P-value ≤ 0.05 was considered statistically significant (red). In the lower heat map (K), the color scale (yellow to blue) highlights an increase in DAG in HFD-fed Low$_{OXPHOS}$ mice. Each sample is a pool from 4 mouse extracts. Histograms show the total amounts of Skm DAG (wt, $n = 8$; Low$_{OXPHOS}$, $n = 8$).

L Blood triglycerides on the 80$^{th}$ day of HFD (wt + HFD, $n = 10$; Low$_{OXPHOS}$ + HFD, $n = 10$).

M Skm relative expression of the GLUT1, GLUT3, PPARβ/δ and PGC1α by qPCR (wt, $n = 5$; Low$_{OXPHOS}$, $n = 5$).

N qPCR relative expression of myokines (wt, $n = 5$; Low$_{OXPHOS}$, $n = 5$). Red or blue color represents higher or lower % expression, respectively, compared to that in wt. SPARC, osteonectin; OSTN, musclin; LIF, interleukin 6 family cytokine; FKN, fractalkine; FSTL1, follistatin-like protein 1; FNDC5, irisin; ADIPOQ, adiponectin.

O Blood glucose following the administration of HFD (wt + HFD, $n = 10$; Low$_{OXPHOS}$ + HFD, $n = 10$). T2D onset occurred on day 80 in wt and day 60 in Low$_{OXPHOS}$ mice.

P Insulin (ITT) and glucose (GTT) tolerance tests on day 60 of HFD (wt + HFD, $n = 10$; Low$_{OXPHOS}$ + HFD, $n = 10$). Low$_{OXPHOS}$ but not wt mice appeared diabetic.

Data information: Bars are the mean ± SEM of the indicated ($n$) mice/genotype *$P < 0.05$ when compared to wt by ANOVA or Student's $t$-test. See also Figs EV3 and EV4, and Table EV3.

ATP synthase inhibition (Figs 4K, and EV5B and C). CII and the ETF complex transfer electrons ($e^−$) from FADH$_2$ to the ETC. Despite this upregulation of CII and ETF proteins (Figs 4K, and EV5B and C), we found an increase in the FAD system protein expression (Fig 4L) and (FADH + FADH$_2$)/FAD ratio (Fig 4M) in Skm from Low$_{OXPHOS}$ mice, suggesting an inefficient oxidation of FADH$_2$ in our system. This could be partially explained by a limited mitochondrial respiration upon ATP synthase inhibition, as well as by the accelerated catabolism of BCAAs (Neinast et al, 2019) in Skm (Figs 1K and 3D, and EV2) and the upregulation of proteins involved in the reduction of FAD to FADH$_2$ (Fig 4L).

**Hindering the ATP synthase activity causes further OXPHOS dysfunction and burst in ROS**

FADH$_2$ accumulation (Fig 4M) may also be a consequence of ETC dysfunction that leads to ROS production (Shadel & Horvath, 2015). To test this possibility, we first assessed CI enzymatic activity in Skm isolated mitochondria. This analysis revealed no significant changes between the two genotypes (Fig EV5D), consistent with the absence of changes in the ETC activity of coupled Skm mitochondria when electrons entered through CI (Fig 1E). We next assayed the activity of CII, the entry point of FADH$_2$ electrons into the ETC. Surprisingly, despite a 2-fold higher expression of CII catalytic subunits (Figs 4K, and EV5B and C) in Low$_{OXPHOS}$ mice, no differences in CII activity were detected between wt and transgenic mice fed with chow or HFD (Fig 5A). These results suggest that CII is not functioning properly in the muscle of Low$_{OXPHOS}$ mice. This is supported by the impaired maximum mitochondrial respiration observed when using CII-specific respiratory substrates, such as succinate or carnitine-palmitate (Fig 5B).

We next sought to understand the mechanisms underlying the inhibition of CII. The activity of several subunits of ETC complexes is regulated by post-translational modifications (Acin-Perez et al, 2011, 2014; Garcia-Bermudez et al, 2018). No changes in the isoelectric point (pI) of the SDHB subunit of CII were detected

between the two genotypes (Fig 5C). However, the acidic shift in the pI of the catalytic subunit SDHA in Low$_{OXPHOS}$ mice compared to that in wt mice (Fig 5D) indicated an altered post-translational modification of the protein that may regulate CII activity (Finley et al, 2011). In particular, Skm from wt animals presented 5 pools of SDHA that differed in the pI (6.5, 6.7, 6.8, 6.9 and 7.1), while only the 2 more acidic pools (pI 6.5 and 6.7) were present in Low$_{OXPHOS}$ mice (Fig 5D). Interestingly, when only the Skm acetyl-K proteins were run into the gel, SDHA was only present in the two more acidic pools (pI 6.5 and 6.7; Fig 5E), suggesting that the basic pools (pI 6.8, 6.9, and 7.1) correspond to deacetylated forms of the protein. Immunoprecipitation of SDHA (revealed with anti-acetyl-K antibody) or acetyl-K proteins (revealed with anti-SDHA antibody) in Skm isolated mitochondria from wt and Low$_{OXPHOS}$ mice confirmed that the protein was more acetylated upon ATP synthase *in vivo* inhibition (Fig 5F). According to previous studies (Finley et al, 2011) and acetylation data from databases [PhosphositePlus®, V6.5.8; (Hornbeck et al, 2015)], at least six acetylation sites on murine SDHA are directly related to CII activity, corresponding to K179 and K182 from the FAD-binding domain and K485, K498, K598, and K608 from the succinate dehydrogenase flavoprotein domain (Fig 5G and H).

CII is also a known site of ROS production (Yankovskaya et al, 2003). In order to demonstrate whether the dysregulation in CII activity is causal for the observed increase in ROS in Low$_{OXPHOS}$ mice, we treated ATPIF1$_{H49K}$ expressing myocytes with specific ETC inhibitors (Fig 5I). As expected, both rotenone and antimycin A, inhibiting CI and CIII respectively, increased ROS production (Fig 5I). However, the treatment of the ATPIF1$_{H49K}$ cultures with malonate, a known inhibitor of CII, reverted ROS amounts to the levels of wt (Fig 5I), indicating a direct participation of CII in the generation of ROS upon ATP synthase inhibition (Fig 5I). Malonate is known to inhibit CII at flavin site (Quinlan et al, 2012). To understand if ROS from CII arose only from the flavin site, we repeated the experiment with carboxin, that inhibits CII at the ubiquinone-binding site (Quinlan et al, 2012). Interestingly, only malonate but

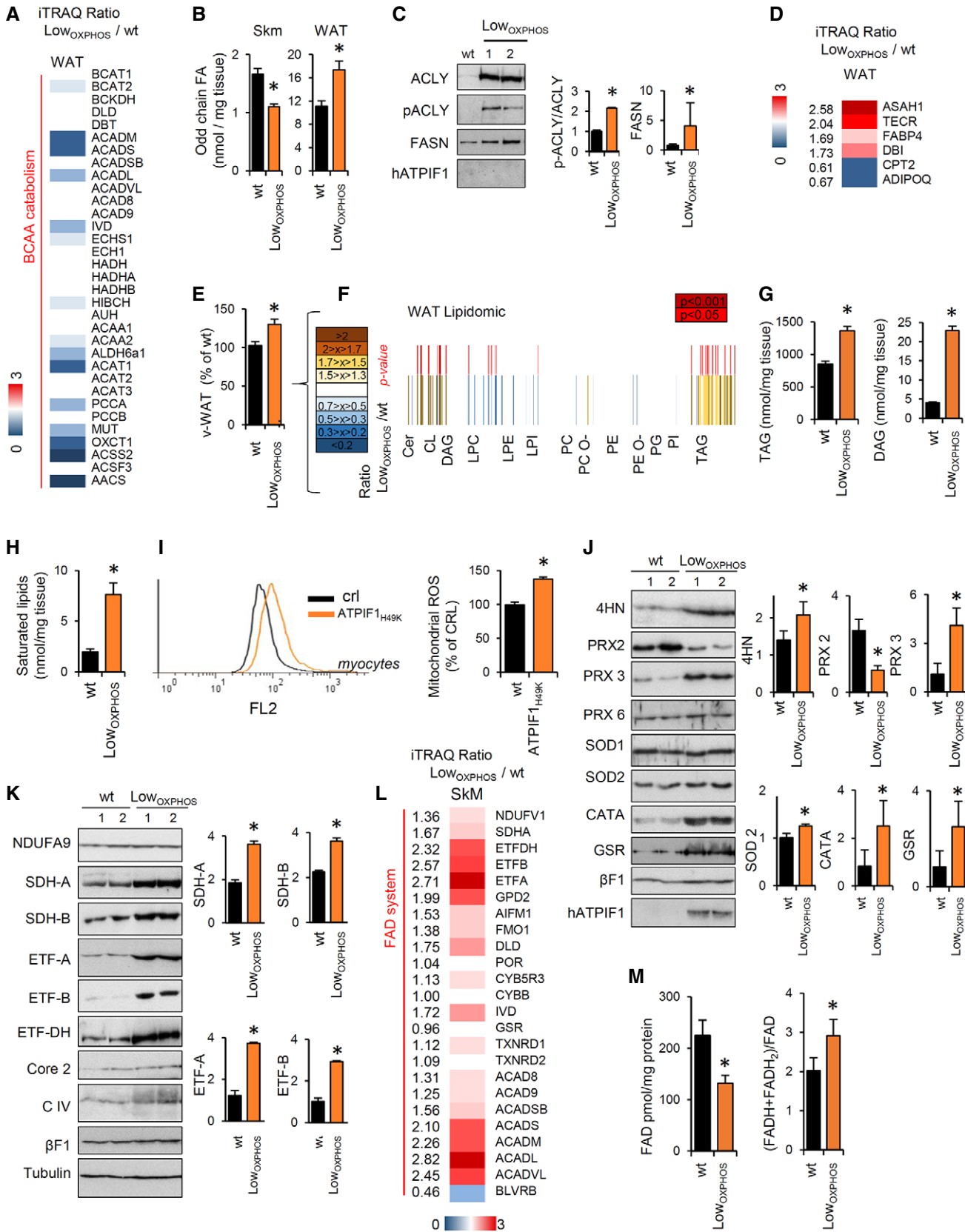

**Figure 4.**

◀

**Figure 4.  The lipogenic switch alters the redox system and lipid-related OXPHOS components.**

Data on wild-type (wt, black bars) and Low$_{OXPHOS}$ (orange bars) mice are shown.

A   v-WAT iTRAQ ratio of proteins from BCAA catabolism (wt, $n = 4$; Low$_{OXPHOS}$, $n = 4$). Higher intensities of blue colors represent lower Low$_{OXPHOS}$/wt expression ratios.

B   Odd chain FAs in Skm and WAT from wt ($n = 8$) or Low$_{OXPHOS}$ ($n = 8$) mice.

C, D   Representative WB expression (C) and iTRAQ analysis (D) of v-WAT proteins from lipid metabolism. Each sample contains extracts from 3 mice. The histogram shows the quantifications (wt, $n = 6$; Low$_{OXPHOS}$, $n = 6$).

E   v-WAT weight expressed as a percentage of the wt (wt, $n = 12$; Low$_{OXPHOS}$, $n = 12$).

F   Quantitative lipidomics in v-WAT. The color scale (brown to blue) in the heat map represents Low$_{OXPHOS}$/wt amounts of specific lipid species. A $P$-value $\leq 0.05$ was considered statistically significant (red). Each sample is a pool from 4 mouse extracts, $n = 8$/genotype.

G   v-WAT amounts of triacylglycerides (TAGs) and diacylglycerides (DAGs) (wt, $n = 8$; Low$_{OXPHOS}$, $n = 8$).

H   Higher saturated lipid species in Low$_{OXPHOS}$ mice compared to wt (wt, $n = 8$; Low$_{OXPHOS}$, $n = 8$).

I   MitoSox staining in myocytes expressing or not ATPIF1$_{H49K}$. The right histogram shows the quantification of mitochondrial ROS. Bars are the mean $\pm$ SEM of $n = 3$ experiments, 12 replicas/condition.

J   Representative WB of Skm lipid peroxidation and redox system proteins. Two samples *per condition* are shown. Each sample contains extracts from 3 mice. Histograms represent quantification (wt, $n = 6$; Low$_{OXPHOS}$, $n = 6$). 4-Hydroxynonenal (4HN), peroxiredoxin 2, 3 and 6 (PRX), superoxide dismutase 1 and 2 (SOD), catalase (CATA) and glutathione reductase (GSR) immunoblots are shown. βF1 is presented as a loading control.

K   Representative WB of Skm OXPHOS and related system proteins. Two samples *per condition* are shown. Each sample contains extracts from 3 mice. Histograms represent quantification (wt, $n = 6$; Low$_{OXPHOS}$, $n = 6$). NDUFA9 (CI), SDHA and B (CII), ETF subunits A and B, ETFDH, CoreII (CIII), subunit 1 (CIV), βF1(CV) and hATPIF1 immunoblots are shown. Tubulin is presented as a loading control. See also Fig EV5.

L   Skm iTRAQ ratio of FADH$_2$-binding proteins (wt, $n = 12$; Low$_{OXPHOS}$, $n = 12$). A higher intensity of red color represents a higher Low$_{OXPHOS}$/wt expression ratio.

M   Oxidized FAD levels and the reduced/oxidized FAD ratio in Skm extracts (wt, $n = 8$; Low$_{OXPHOS}$, $n = 8$).

Data information: Bars are the mean $\pm$ SEM of the indicated ($n$) mice/genotype. *$P < 0.05$ when compared to wt by Student's *t*-test. See also Figs EV4 and EV5, and Table EV3.

Source data are available online for this figure.

not carboxin prevented the ATPIF1-dependent ROS production (Fig 5J), suggesting a specific role for SDHA in this event.

Building upon the observation that high levels of ROS may impair the stability and activity of ETC complexes (Acin-Perez *et al*, 2008; Cogliati *et al*, 2013), we next tested the effect of ATPIF1$_{H49K}$ on ETC complex superassembly (Fig 5K). Blue-native (BN) gels and relative clear-native (CN) *in-gel* activities were performed on IMM solubilized proteins from wt or Low$_{OXPHOS}$ hindlimb muscles (Fig 5K). No significant alterations in functionality and supramolecular organization of CII, CIII, and CIV of the ETC were observed (Fig 5K), whereas consistent with previous reports (Santacatterina *et al*, 2016), the presence of ATPIF1$_{H49K}$ increased the oligomerization of CV (Fig 5K). Remarkably, in Low$_{OXPHOS}$ mice, a subassembly of CI with no activity was observed (Fig 5K), suggesting an early phase of ROS-mediated CI degradation (Guaras *et al*, 2016).

### Edaravone increases mitochondrial function and FA β-oxidation

In order to identify an activator of mitochondrial function that may restore Skm homeostasis, we tested the effect of 702 FDA-approved drugs on O$_2$ consumption in C$_2$C$_{12}$ myocytes using palmitate as a substrate (Fig 6A). This approach identified 41 compounds with a potential mitochondrial enhancer function, as defined by an increased maximum respiration (Fig 6A and B).

Next, we analyzed the effect of the identified hits on 9,10$^3$H(N)-palmitic acid oxidation in myocytes (Fig 6C), a readout of mitochondrial FFA β-oxidation. This analysis uncovered edaravone, a lipophilic antioxidant drug currently used to recover from stroke in Japan and to treat ALS in the US (Rothstein, 2017), as a potent activator of β-oxidation (Fig 6D). Edaravone treatment exhibited the capacity to increase basal, oligomycin-sensitive, and maximum mitochondrial respiration (Fig 6E). Moreover, edaravone partially restored palmitate β-oxidation flux inhibited by ATPIF1$_{H49K4}$ in C$_2$C$_{12}$ (Fig 6F) and in primary myocytes from Low$_{OXPHOS}$ mice (Fig 6G).

### *In vivo* edaravone treatment restores ROS imbalance and lipid metabolism

In order to understand whether the previously reported antioxidant property of edaravone (Rothstein, 2017) is mitochondria-mediated, we assayed its capacity of quenching mitochondrial ROS in myocytes. Confirming the hypothesis, 3-h treatment with edaravone or mitochondrial-specific antioxidant MitoQ but not N-acetyl-cysteine (NAC) significantly reduced mitochondrial ROS production (Fig 7A). To increase mitochondrial activity and reduce ROS in Skm *in vivo*, we administered 3 mg/kg edaravone to wt and Low$_{OXPHOS}$ littermates (Fig 7B–L). Consistent with its *in vitro* effect, a 1-month edaravone administration reversed the ATPIF1$_{H49K}$-dependent alterations in the redox system (Fig 7B).

We next tested the effect of reverting the mitochondrial imbalance observed in Low$_{OXPHOS}$ mice on hyperlipidemia and insulin resistance. Indeed, mitochondrial ROS are known modulators of lipogenesis through the Akt-PGC1α axis (Martinez-Reyes & Cuezva, 2014). Accordingly, edaravone and MitoQ but not NAC significantly increased the myocyte expression of PGC1α and reduced the TNFα production (Fig 7C). Consistently and in line with the observed edaravone-mediated increase in lipid catabolism *in vitro* (Fig 6F and G), the levels of the rate-limiting lipid synthesis enzymes ACLY and FASN were downregulated in both wt and Low$_{OXPHOS}$ mice administered edaravone (Fig 7D). Remarkably, in Low$_{OXPHOS}$ animals, a 1-month edaravone treatment restored the levels and activity of mitochondrial OXPHOS complexes. Thus, ETFA, ETFB, and SDHA protein expression was reduced to wt levels (Fig 7E). Similarly, edaravone restored the physiological level of acetylation of SDHA (CII) (Fig 7F) and abolished the ROS-mediated degradation of CI (Fig 7G) observed in Low$_{OXPHOS}$ mice (Fig 5K). As a consequence of these events, when mice were fed a HFD and simultaneously treated with edaravone, no differences were observed in body weight (Fig 7H), Skm whitening (Fig EV5E), or in v-WAT (Fig 7I) between

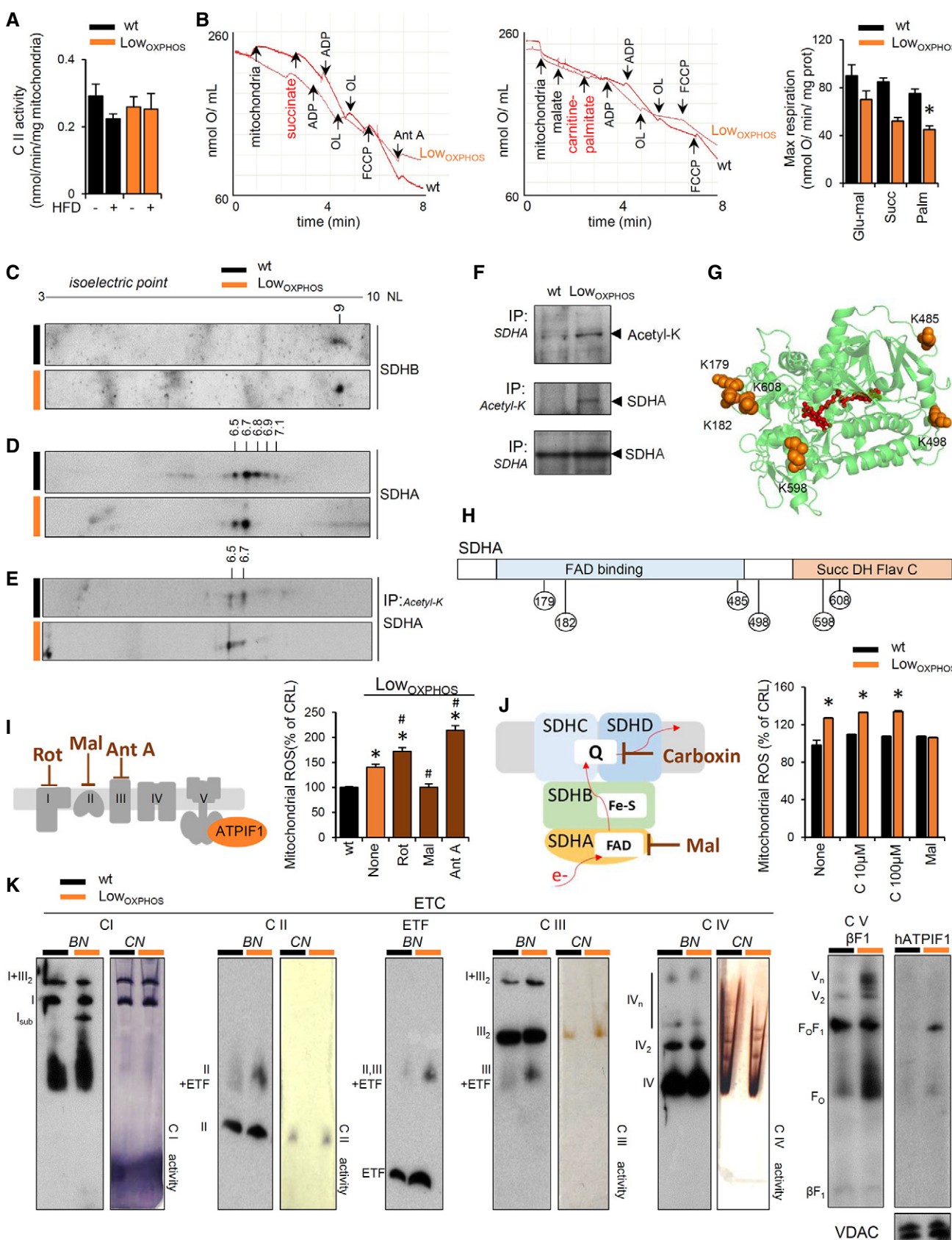

**Figure 5.**

**Figure 5.   Hindering the ATP synthase activity feeds back to OXPHOS dysfunction by inhibiting CII and generating a SDHA-dependent burst in ROS.**

A   Enzymatic activity of respiratory complex CII in Skm isolated mitochondria from wt and Low$_{OXPHOS}$ mice fed with chow or HFD (wt, $n = 4$; Low$_{OXPHOS}$, $n = 4$; wt + HFD, $n = 4$; Low$_{OXPHOS}$ + HFD, $n = 4$).

B   Polarographic profiles of isolated mitochondria from wt (lower trace) and Low$_{OXPHOS}$ (upper trace) animals using succinate (left graph) or palmitoyl-carnitine (right graph) as a substrate. Quantification of maximal respiration in the right histogram. Bars are the mean ± SEM of $n = 3$ mice/genotype, 3 traces/mouse; OL, oligomycin; Ant A, antimycin A.

C–E   2D-PAGE of Skm extracts (C, D) and acetylated proteins (E) from wt ($n = 4$) and Low$_{OXPHOS}$ ($n = 3$) mice. The isoelectric point (pI) of the CII subunits SDHB (C) and SDHA (D) was calculated by protein migration in pH 3–10 NL strips.

F   Immunocapture (IP) of SDHA blotted with anti-acetyl-K antibody (upper panel) and of acetylated proteins blotted with anti-SDHA antibody (middle panel) in Skm isolated mitochondria from wt ($n = 3$) and Low$_{OXPHOS}$ ($n = 3$) mice.

G, H   PyMOL cartoon representations of acetylated lysines in SDHA related to CII activity. Six acetylated-K are shown.

I   MitoSox staining in myocytes expressing or not ATPIF1$_{H49K}$. The left scheme illustrates where each ETC inhibitor works. The right histogram shows the quantification of mitochondrial ROS. Bars are the mean ± SEM of $n = 3$ experiments, 12 replicas/condition.

J   MitoSox staining in myocytes expressing or not ATPIF1$_{H49K}$. The left scheme illustrates where each CII inhibitor works. The right histogram shows the quantification of mitochondrial ROS. Bars are the mean ± SEM of $n = 3$ experiments, 12 replicas/condition.

K    Representative blue-native immunoblots (BN) and clear-native *in-gel* activity (CN) of Skm mitochondrial membrane proteins from wt ($n = 3$) and Low$_{OXPHOS}$ ($n = 3$) 2-month-old mice. The migration of the respiratory complexes/supercomplexes CIV is indicated. Subassembly of CI (with no activity, suggesting degradation) was observed in Low$_{OXPHOS}$ mice. VDAC is shown as a loading control.

Data information: \*,#$P < 0.05$ when compared to wt or Low$_{OXPHOS}$, respectively, by ANOVA and Student's *t*-test. See also Fig EV5.
Source data are available online for this figure.

Low$_{OXPHOS}$ and wt mice. Moreover, in Low$_{OXPHOS}$ mice a slight reduction in plasma BCAA levels (Fig EV5F) and 14C(u)-BCCA uptake (Fig EV5G) was observed compared to no-treated animals. Despite edaravone did not significantly affect food intake (Fig EV5H), at day 60 of HFD, when nontreated Low$_{OXPHOS}$ mice were already prediabetic, edaravone-treated Low$_{OXPHOS}$ mice showed similar insulin and glucose sensitivity to control mice (Fig 7J). To note that edaravone delayed the onset of T2D in both genotypes: Interestingly, at day 80 of HFD, when both nontreated wt and Low$_{OXPHOS}$ mice were insulin-resistant (Fig 7K), GTT values indicated that edaravone-treated animals were still insulin-sensitive (Fig 7K) and developed T2D only at day 90–100 of HFD (Fig 7K and L). Altogether, these *in vivo* studies identify edaravone as a drug with therapeutic potential for the treatment of mitochondrial-derived metabolic disorders.

## Discussion

Increasing evidence has associated muscle mitochondrial dysfunction with metabolic syndrome, but whether the former is a cause or a consequence of the latter remains controversial. To answer this question, we generated the first mouse model of the interference of mitochondrial activity in Skm by the expression of the human active inhibitor of ATP synthase (Formentini et al, 2014; Boreikaite et al, 2019) in ACTA1-positive myocytes. These animals presented higher fat deposits in both Skm and WAT, indicating perturbations in whole-body lipid metabolism. In oxidative Skm, reciprocal cross-talk between glucose and FA metabolism determines fuel selection through the Randle cycle (Samuel & Shulman, 2016). Altered nutrient utilization and metabolic rewiring upon defective mitochondrial activity has been proposed with different pathway fluxes involved (Gaude & Frezza, 2014; Liu et al, 2016; Gaude et al, 2018). Consistent with this, in our model, impairment of OXPHOS triggers the shift to a lipogenic phenotype of a highly oxidative nonlipogenic tissue such as muscle (Fig 8). Indeed, upon ATP synthase inhibition, lipogenic enzymes are overexpressed and activated, and acetyl-CoA accumulates in myofibers to fuel the rate of *de novo* lipid

synthesis. In physiological and normoxic conditions, glycolysis and FFA β-oxidation represent the major sources of cellular acetyl-CoA (Pietrocola et al, 2015). However, the contribution of these two pathways to promoting lipogenesis in our model is unlikely due to the phosphorylation-mediated inactivation of PDH that reroute the glycolytic rate to lactate production (Patel et al, 2014) along with the observed inhibition of FFA β-oxidation in Low$_{OXPHOS}$ mice. In this regard, we suggest that a reduction in the FFA entrance into the mitochondria (Foster, 2012; Formentini et al, 2017a) along with the acetyl-CoA-mediated acetylation and inactivation of proteins involved in β-oxidation (Pietrocola et al, 2015) may play a part in the mechanism underlying the reduction in lipid catabolism upon ATP synthase inhibition.

We are also tempted to exclude that the generated lactate was transformed into glycerol-3-phosphate and utilized for lipid synthesis. Despite PCK1, the master regulator of glyceroneogenesis (Hanson & Hakimi, 2008), is highly overexpressed in Skm of Low$_{OXPHOS}$ mice (Fig EV1), glyceroneogenesis is a minor pathway in muscle, and the carbon$^{14}$ atoms derived from labeled glucose accumulate in lipids in a similar manner in control and ATPIF1$_{H49K}$ expressing myocytes.

An alternative route for acetyl-CoA production is through utilization of BCAAs (White & Newgard, 2019). This pathway shares with β-oxidation many enzymes that are highly upregulated in our proteomic dataset in Low$_{OXPHOS}$ mice. Interestingly, the ATP synthase-mediated LD formation in myocytes is prevented by removing BCAAs from culture media, what might suggest that upon OXPHOS inhibition, BCAA oxidation participates in acetyl-coA synthesis and increased lipogenesis. Supporting this hypothesis, carbon$^{14}$ leucine and isoleucine are highly oxidized to $CO_2$ and accumulated faster in myocyte lipid fraction when the ATP synthase is inhibited. Consistently, it has been proposed that upon metabolic stress, BCAA catabolism fuels lipogenesis (Green et al, 2016; Wallace et al, 2018) and competes for lipid oxidation, causing the accumulation of toxic lipid intermediates (White et al, 2016). Insulin-resistant mice have been described to shift BCAA metabolism from WAT and liver toward muscle, increasing the Skm rate of BCAA oxidation (Neinast et al, 2019), what is observed in our

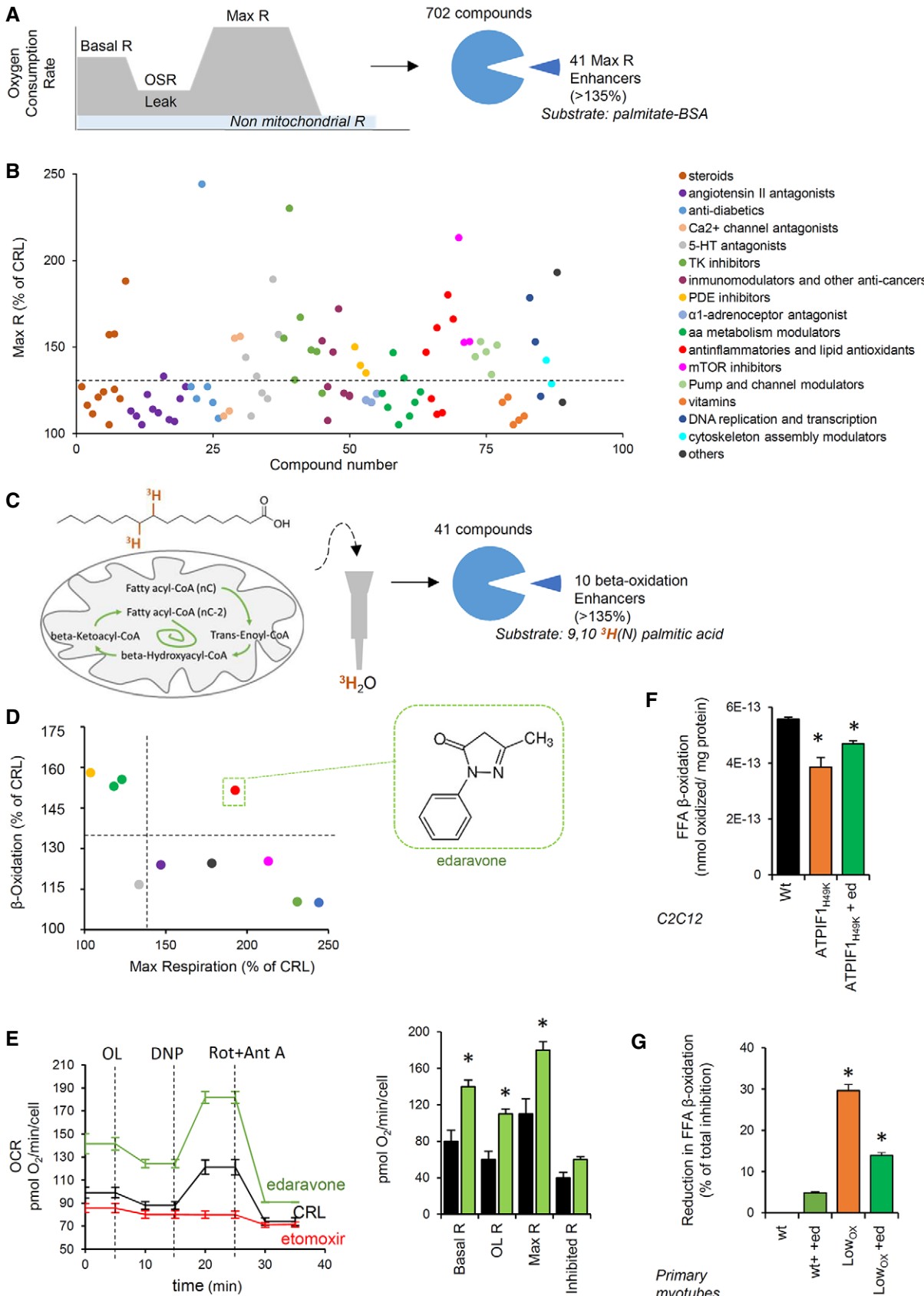

Figure 6.

◀

**Figure 6. Edaravone as a mitochondrial metabolism enhancer.**

A, B  The schematic illustrates the screening process of 702 drugs related to the mitochondrial respiratory capacity assessed by Seahorse XFe96 in $C_2C_{12}$ myocytes. Palmitate is used as a substrate. A total of 41 hits were identified. Maximum respiration (Max R, % of untreated cells) of compounds categorized by their therapeutic use (B).

C, D  The schematic illustrates the screening process of the 41 hits on FFA β-oxidation using $9,10^3H(N)$ palmitic acid as a substrate in $C_2C_{12}$ myocytes. Ten enhancers were identified. (D) Correlation between the maximum respiration (x) and palmitate β-oxidation (y). Colors identify classes of pharmaceuticals as in (B). Edaravone was selected as a hit.

E  Representative respiratory profile of $C_2C_{12}$ myocytes treated (green trace) or not treated (black trace) with 2 μM edaravone. The CPT1 inhibitor etomoxir was used as a negative control. OCR, oxygen consumption rate; OL, oligomycin; DNP, 2,4-dinitrophenol; Rot, rotenone; Ant A, antimycin A. Quantification in right histogram. Bars are the mean ± SEM of 12 replicas/condition.

F, G  FFA β-oxidation in $C_2C_{12}$ myocytes transfected with CRL or ATPIF1$_{H49K}$ plasmids (F) and in primary myotubes from wt and Low$_{OXPHOS}$ mice (G), treated (green bars) or not with 2 μM edaravone. Bars are the mean ± SEM of $n$ = 3 experiments, 9 replicas/condition.

Data information: (E–G) $*P < 0.05$ when compared to wt by ANOVA and Student's *t*-test.

model. This also results in faster lipid transport across the membranes favoring higher FFA blood levels and lipotoxicity (Jang *et al*, 2016). Accordingly, acyl-glyceride species accumulated in the myofibers and blood of Low$_{OXPHOS}$ mice, which caused these animals to develop T2D faster than control animals. Indeed, the application of metabolomics technologies has revealed that BCAA fluxes and related metabolites are more strongly associated with homeostatic model assessment of insulin resistance (HOMA-IR) than many common lipid species (Newgard, 2012). Remarkably, metformin, the first-line medication for the treatment of T2D and known OXPHOS modulator, has been shown to significantly diminish BCAA oxidation (Liu *et al*, 2016).

We cannot exclude the contribution of another minor pathway to the observed lipogenesis because acetyl-CoA could be partially derived from the rewiring of glutamine metabolism through reductive carboxylation (Mullen *et al*, 2011). This is a form of rewired TCA cycle that in Skm may occur under specific metabolic stresses, such as hypoxia (Metallo *et al*, 2011) or mtDNA mutations (Chen *et al*, 2018; Gaude *et al*, 2018), and produces acetyl-coA from glutamine, prompting the anaplerotic reactions that increase metabolite levels for lipid synthesis. Interestingly, iTRAQ analysis revealed an upregulation of the key regulators of reductive carboxylation IDH (Metallo *et al*, 2011) and MDH (Gaude *et al*, 2018) in Skm from Low$_{OXPHOS}$ compared to levels in wt mice.

Although the implication of BCAA metabolism and reductive carboxylation in the setting of IR is still object of debate, the connection between the ectopic accumulation of lipids and impaired glycemic control has been widely documented (Despres & Lemieux, 2006). Both ceramide and DAG accumulation have been reported in Skm and liver from T2D subjects (Szendroedi *et al*, 2014), as well as a direct connection between lipotoxicity and IR (Jornayvaz & Shulman, 2012; Turpin-Nolan *et al*, 2019). Specifically, DAG species containing C16:0, C18:0, C18:1, C18:2, or C20:4 FA showed the strongest relationship with IR in obese and T2D individuals (Szendroedi *et al*, 2014). Interestingly, changes in these species have been detected in our model (Figs 3K, and EV4D and E). However, a muscle-specific phosphoethanolamine cytidylyltransferase knockout mouse, displaying a 200% increase in Skm accumulation of DAGs but normal mitochondrial β-oxidation activity, failed to display impaired insulin sensitivity and presented unexpectedly enhanced exercise performance (Selathurai *et al*, 2015). This could be related to the capacity of β-oxidation to remove fat from muscles, circumventing ectopic lipid accumulation, and resulting in increased energy production. Accordingly, we suggest that neither

mitochondrial dysfunction nor perturbations in lipid species *per se* are determinants for the pathological setting, pointing to the necessary combination of the two events for IR. Supporting this hypothesis, a modification of the hepatic sphingolipid pattern that finally results in the impairment in mitochondrial homeostasis has been recently demonstrated to reduce insulin sensitivity (Hammerschmidt *et al*, 2019). Interestingly, similar results are observed in our complementary model, where mitochondrial activity inhibition triggers an alteration in the abundance of several lipid species, including ceramides, DAGs, and TAGs.

TAG synthesis in particular has been recently linked to mitochondrial activity (Benador *et al*, 2018). The cross-talk between lipid storage and mitochondria in fat-oxidizing tissues such as brown adipose tissue (BAT) has been described to facilitate acyl-glyceride production and lipid vesicle expansion (Benador *et al*, 2018, 2019). In line with this, we report that Skm intrafiber formation of LDs occurred in an ATP synthase-dependent manner, which may explain the observed burst in lipid production and storage upon OXPHOS inhibition. Accordingly, liver-specific ablation of MFN-2, which triggers defective mitochondrial fusion, modified PS transfer and phospholipid synthesis, and leads to the development of NASH-like hyperlipidemia (Hernandez-Alvarez *et al*, 2019). Therefore, herein we support the idea that mitochondrial OXPHOS is crucial in maintaining lipid and glucose homeostasis.

Mechanistically, we propose that the ATP synthase-dependent dysregulation of Skm lipids further promotes a dysfunctional ETC that results in boosting ROS production. Upon the increase in lipogenesis, the switch from NADH to FADH$_2$ leads to a greater proportion of electrons being transferred to CII or other FAD-dependent dehydrogenases rather than to CI. When FADH$_2$ is used as a substrate for mitochondrial respiration, $O_2$ consumption increases to maintain membrane potential (Hinkle, 2005) and favors the so-called reversed electron transfer (Scialo *et al*, 2016). Under these conditions, superoxide production is enhanced (Robb *et al*, 2018), which contributes to CI degradation (Guaras *et al*, 2016). In our model, the acetylation-mediated partial inhibition (Finley *et al*, 2011) of the respiratory CII catalytic subunit SDHA, along with the overexpression of FADH$_2$-producing enzymes, triggers the accumulation of FADH$_2$ in myocytes.

Inhibiting the activity of ATP synthase is known to stimulate the production of ROS (Balaban *et al*, 2005; Martinez-Reyes & Cuezva, 2014). This mild ROS signal is not detrimental in terms of cell death and rather seems to participate in an ATP synthase-mediated signal transduction to the nucleus (Formentini *et al*, 2012). However, no

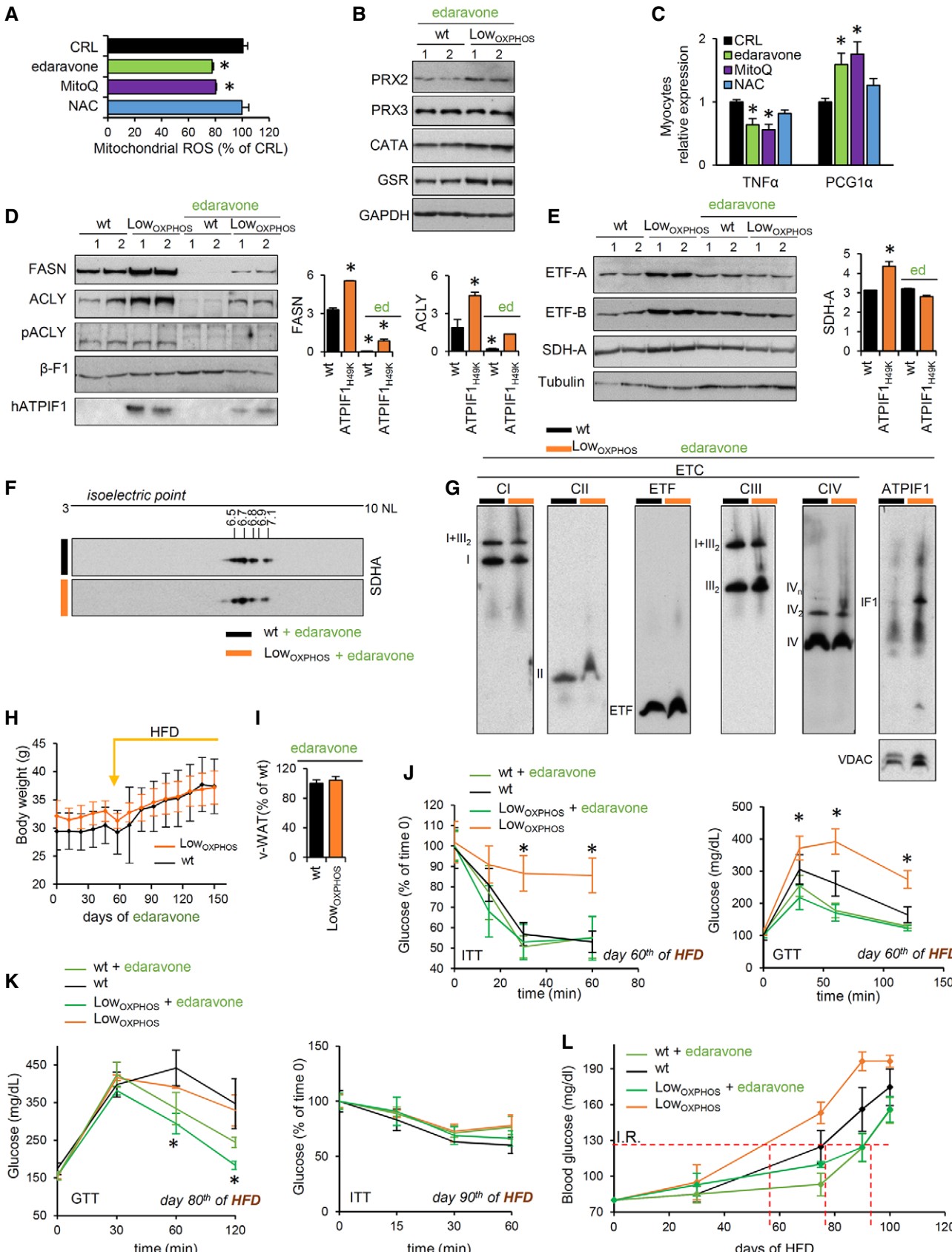

**Figure 7.**

◄

**Figure 7.** *In vivo* edaravone treatment rewires the ROS-driven lipogenic switch and prevents IR.

Data on wild-type (wt, black bars and traces) and Low$_{OXPHOS}$ (orange bars and traces) mice treated or not with 3 mg/kg edaravone are shown.

A Mitochondrial ROS after 3 h treatment with 2 µM edaravone (green bars), 10 nM MitoQ (purple bars) or 1 mM NAC (blue bars). Bars are the mean ± SEM of $n = 3$ experiments, 9 replicas/condition.

B Representative WB expression of Skm ROS system proteins in mice administered (30 days) edaravone. Two samples *per condition* are shown. Each sample contains extracts from 3 mice (wt + edaravone, $n = 6$; Low$_{OXPHOS}$ + edaravone $n = 6$). Peroxiredoxin 2 and 3 (PRX), catalase (CATA) and glutathione reductase (GSR) immunoblots are shown. GAPDH is presented as a loading control.

C qPCR relative expression of TNFα and PGC1α in myocytes treated for 24 h with 2 µM edaravone (green bars), 10 nM MitoQ (purple bars) or 1 mM NAC (blue bars). Six replicas/condition.

D, E Representative WB expression of Skm *de novo* lipid synthesis (D) and OXPHOS (E) proteins. One month of edaravone treatment downregulated FASN and ACLY in both wt and Low$_{OXPHOS}$ mice (D) and rewired the upregulation of ETF-A, ETF-B and SDHA proteins to the levels of wt (E). Two samples *per condition* are shown. Each sample contains extracts from 3 mice. Tubulin or βF1 is presented as a loading control. Quantifications in lateral histograms (wt, $n = 6$; Low$_{OXPHOS}$, $n = 6$; wt + edaravone, $n = 6$; Low$_{OXPHOS}$ + edaravone, $n = 6$).

F 2D-PAGE of Skm extracts from wt ($n = 3$) and Low$_{OXPHOS}$ ($n = 3$) mice treated with edaravone. The pI of SDHA, calculated by protein migration in pH 3-10 NL strips, was the same for both genotypes.

G Representative blue-native immunoblots (BN) of Skm mitochondrial membrane proteins from 2-month-old mice treated with edaravone for 30 days (wt + edaravone, $n = 3$; Low$_{OXPHOS}$ + edaravone, $n = 3$). The migration of the respiratory complexes/supercomplexes CI-CIV and hATPIF1 is indicated. VDAC is shown as a loading control.

H Mouse body weight following the administration of HFD and edaravone.

I v-WAT amounts at day 60$^{th}$ of HFD in mice treated with edaravone.

J Insulin (ITT) and glucose (GTT) tolerance tests after 60 days of HFD in mice treated with edaravone.

K GTT after 80 days of HFD (left) and ITT after 90 days of HFD (right) in wt and Low$_{OXPHOS}$ mice treated with edaravone, showing that the compound has an improving effect itself.

L Blood glucose following the administration of HFD. T2D onset occurred on day 90 in wt and Low$_{OXPHOS}$ mice treated with edaravone.

Data information: H–L: wt + HFD, $n = 5$; Low$_{OXPHOS}$ + HFD $n = 4$; wt + HFD + edaravone, $n = 10$; Low$_{OXPHOS}$ + HFD + edaravone $n = 9$. Bars are the mean ± SEM of the indicated ($n$) mice/genotype. *$P < 0.05$ when compared to wt by ANOVA and Student's *t*-test. See also Fig EV5.

Source data are available online for this figure.

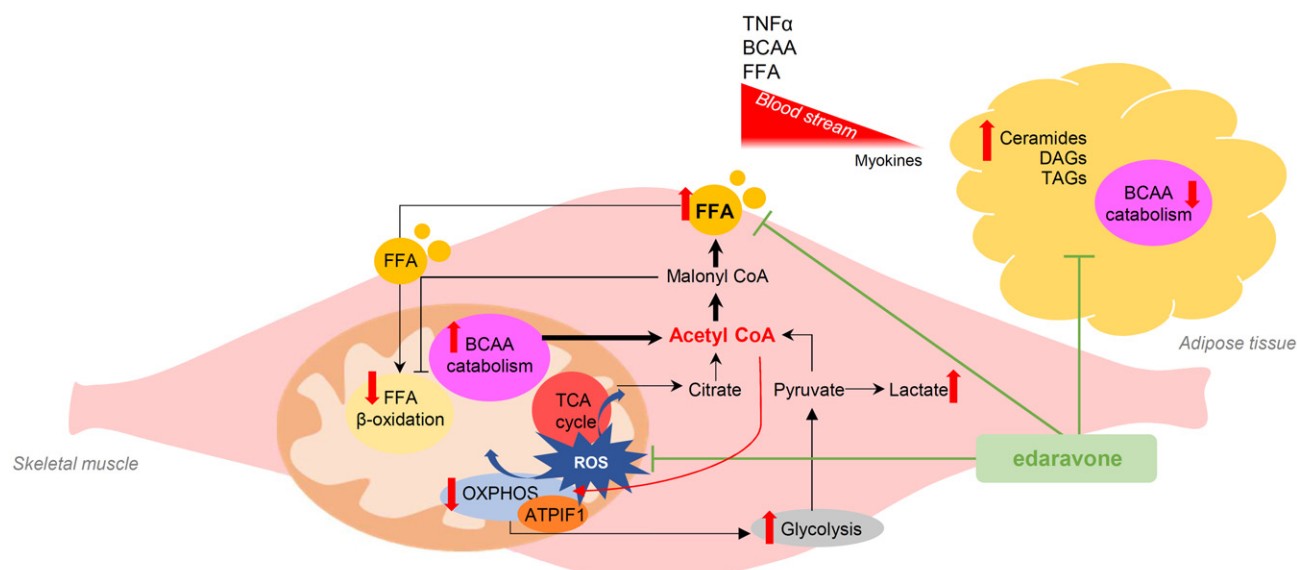

**Figure 8.** The OXPHOS-mediated lipogenic reprogramming is prevented by edaravone.

The schematic illustrates the effect of *in vivo* edaravone administration in Skm from Low$_{OXPHOS}$ mice. Restraining OXPHOS generates a ROS-mediated rewiring of lipid metabolism through enhanced BCAA catabolism and lipid synthesis. Skm and WAT perturbations in lipid species are observed, defining a Low$_{OXPHOS}$ phenotype of adiposity and lipotoxicity. Edaravone treatment restores normal ROS and lipid metabolism, reducing v-WAT deposits and preventing the setting of IR mediated by mitochondrial dysfunction.

sites for ROS production have been described inside the ATP synthase structure (Zhao *et al*, 2019). We herein demonstrated that the acetylation and inhibition of SDHA is causal in the ATP synthase-mediated boost in ROS, which finally triggers a metabolic response in mice that affects IR. In order to support this, we provide evidence that edaravone supplementation, which induced a ROS-mediated repression of lipogenesis, reestablished lipid homeostasis and insulin sensitivity (Fig 8). Although the antioxidant property of edaravone is known (Rothstein, 2017), the exact mechanism of action for this drug is unknown. Herein, we unveiled its effect on

targeting mitochondria, inhibiting superoxide production and inducing a hypolipidemic rewiring of mitochondrial lipid metabolism to an increase FA catabolism.

Altogether, this study provides a mechanistic explanation for the link between mitochondrial dysfunction and the onset of metabolic disorders. The clinical relevance of our preclinical findings should be validated before application to human health. Nevertheless, our data suggest edaravone as a potential drug of interest for a novel approach in mitochondrial-derived metabolic disorders. Furthermore, the fact that edaravone is a FDA-approved molecule may guarantee a rapid translation into clinical applications.

# Materials and Methods

### Reagents

The 11,018 FDA-Approved Library (Selleckchem) was used; $C_2C_{12}$ (ATCC©) or primary myocytes were incubated for 3 h with 702 compounds from the library at a final concentration of 2 μM. Edaravone was purchased from Selleckchem and used at 2 μM (*in vitro*) or injected i.p. at 3 mg/kg (*in vivo*). MitoQ was kindly gifted by Professor Michel P. Murphy. A comprehensive list of reagents and antibodies utilized is detailed in Table EV1.

### Animal studies

All animal experimentation procedures were performed after the approval of the Institutional Review Board (Ethical Committee of the UAM University and Madrid Community, Spain; CEI-24-571, PROEX 183/17) in compliance with animal policies and ethical guidelines of the "European Guidelines for the Care and Use of Laboratory Animals" (EU Directive 86/609). All procedures follow the ARRIVE guidelines.

For the *in vivo* studies, B6;C3-Tg(ACTA1-rtTA,tetO-cre)102MonK/J mice (ATPIF1$_{H49K}$|$^T$) were purchased from The Jackson Laboratories. The Tet-on double transgenic ATPIF1$_{H49K}$|$^{T/H}$ mouse was obtained by breeding ATPIF1$_{H49K}$|$^T$ with the ATPIF1$_{H49K}$|$^H$ mouse (Formentini *et al*, 2014), which integrates in its genome the ATPIF1$_{H49K}$-TRE construct under a tetracycline-regulated promoter. Mice were maintained on the (C57BL/6 × C3H)F2 background. Administration of 2 mg/ml doxycycline in the drinking water for at least 2 weeks was used to turn on the Skm expression of ATPIF1$_{H49K}$ protein. All experiments were performed on age-matched male littermate wt and ATPIF1$_{H49K}$|$^{T/H}$ mice. In order to minimize the number of animals, we used power analysis to calculate the minimum sample size using the free software DOEUMH (https://samplesizeumh.shinyapps.io/DOEUMH) based on the TrialSize library of the R program (R Core Team). We selected the procedure KMeans—ANOVA, fixing the significance to 0.05, power to 0.08, and a drop-out of 5%. We took into consideration differences between averages of about 1.5, as for omic studies. Minimum number of mice/group is 12 mice/group. Mouse motor function was evaluated by rotarod, grip force/strength, and open field tests. Details are shown in the Expanded View Section. All tests were performed in a blinded fashion. Randomization was assessed by equally distributing experimental groups across multiple cages and balancing the location of the mouse cages on the racks.

### T2D onset and monitoring: glucose and insulin tolerance tests

Eight-week-old male animals were fed *ad libitum* either a standard or a HFD diet (Research Diets) for 14 weeks. To monitor the onset of T2D, blood glucose and body weight were measured once and twice a week, respectively. Glucose and insulin tolerance tests (GTT and ITT) were performed by injecting glucose (0.2 g/mg) or insulin (0.8 UI/kg) after 12 or 4 h of starvation, respectively. The blood glucose concentration was measured using One Touch Select Plus strips (Johnson & Johnson) and a measurement apparatus.

### Primary cultures of myotubes

Primary Skm cell cultures (myotubes) were established from hindlimb muscle biopsies from wt or ATPIF1$_{H49K}$|$^{T/H}$ mice (Formentini *et al*, 2017a). Cells were propagated in SkGM media (Lonza) and supplemented with the accompanying bullet kit but omitting insulin. At 80–90% confluence, cells were differentiated in α-MEM containing 2% FBS and 100 nM insulin. $C_2C_{12}$ mouse myoblasts were cultured in DMEM 10% FBS, 1 mM glutamine, and amino acids. At 80–90% confluence, cells were differentiated in DMEM containing 2% FBS and 100 nM insulin.

### Cell transfection

At ~70% confluence, myoblasts were transfected with CRL or ATPIF1$_{H49K}$ plasmid (Formentini *et al*, 2017a) (pCMV-SPORT6-ATPIF1$_{H49K}$ or pCMV-SPORT6-control) using Lipofectamine 2000 transfection reagent. Experiments were performed 24 h post-transfection.

### Skeletal muscle mitochondria

Fresh hindlimb muscles from wt and ATPIF1$_{H49K}$|$^{T/H}$ mice were minced and homogenized in a glass–glass homogenizer in buffer A (320 mM sucrose, 1 mM EDTA, and 10 mM Tris–HCl, pH 7.4). Skm mitochondria were obtained by centrifugation as previously described (Formentini *et al*, 2017a). Briefly, unbroken cells and tissue were removed by centrifugation at 1,000 *g* for 5 min at 4°C; mitochondria were obtained by supernatant centrifugation at 11,000 *g* for 15 min at 4°C.

### Oxygen consumption rates

The oxygen consumption rate (OCR) in isolated mitochondria (200 μg protein) was determined with a Clark-type electrode. Glutamate/malate (10 mM), succinate (10 mM), or malate (2 mM) plus palmitoyl-carnitine (0.05 mM) was used as respiratory substrates in the presence or absence of 0.5 mM ADP, 5 μM oligomycin (OL), 5 μM FCCP, and 1 μM antimycin A (Ant A). The composition of the respiration buffer is 225 mM sucrose, 5 mM MgCl$_2$, 10 mM KCl, 10 mM phosphate buffer, 1 mM EGTA, 0.05% BSA, and 10 mM Tris–HCl, pH 7.4.

The OCR in myotubes and $C_2C_{12}$ cells treated with the drug library was determined in an XF96 Extracellular Flux Analyzer with the XFe96 Flux Pack following the manufacturer's protocols. Cells were starved for 12 h in low-glucose DMEM (0.05 mM glucose, 1% FBS) and then changed to KHB media (111 mM NaCl, 4.7 mM KCl, 1.25 mM glutamine, 5 mM HEPES, pH 7.4). BSA-conjugated

palmitate (1 mM sodium palmitate, 0.17 mM BSA solution) was added as the main substrate. The final concentration and order of injected substances was 3 μM OL, 0.25 mM DNP, 1 μM rotenone, and 1 μM antimycin A. When indicated, 1 mM carnitine, 10 nM etomoxir, or 2 μM of FDA-approved library compounds were added. Compounds that positively affected respiration by a factor of at least 30% were selected for further screenings.

### Determination of mitochondrial enzyme activities

Isolated mitochondria were used for the spectrophotometric determination of respiratory complexes I and II, according to Barrientos *et al* (2009) with minor modifications. Complex I activity was measured at $A_{340}$ using 100 μg of mitochondria. Mitochondria were resuspended in 1 ml of CI/CII buffer (25 mM $K_2HPO_4$, 5 mM $MgCl_2$, 3 mM KCN, and 2.5 mg/ml bovine serum albumin) containing 0.1 mM $UQ_1$, 0.1 M NADH, and 1 mg/ml antimycin A. Inhibition of the activity was accomplished by the addition of 1 μM rotenone. Complex II activity was measured at $A_{600}$ using 100 μg of mitochondria. Mitochondria were resuspended in 1 ml of CI/CII buffer containing 30 μM DCPIP, 1 μM rotenone, 1 μM antimycin A, 10 mM succinate, and 6 mM phenazine methosulfate.

### Mitochondrial ROS

The mitochondrial production of superoxide in myocytes was monitored by flow cytometry using 5 μM MitoSoX (Formentini *et al*, 2012). Cells were analyzed in a BD FACScan. Where indicated, ETC inhibitors were added at following concentrations: 1 μM rotenone, 100 μM malonate, 10 or 100 μM carboxin, 1 μM antimycin A.

### FAD, ATP, NADP, BCAA, lactate, FFA, free glycerol, and acetyl-CoA determinations

Metabolites were measured in 40 mg of hindlimb muscle or adipose tissue from wt and ATPIF1$_{H49K}$|$^{T/H}$ mice using a FLUOstar-Omega spectrophotometer (BMG LABTECH).

Tissue was homogenized in 1 M perchloric acid, and the oxidized FAD content was measured colorimetrically in neutralized samples using the FAD Assay Kit (Abcam) at $A_{570}$. Phenazine methosulfate was used on neutralized samples to transform free FADH and $FADH_2$ into FAD and measure the total pool of FAD + FADH + FADH2. The ATP concentration in muscle was determined using the ATP Bioluminescence Assay Kit CLS II (Roche). NADP/NADPH Quantification Kit (Sigma-Aldrich) and PicoProbe Acetyl-CoA Fluorometric Assay Kit (BioVision) were assessed following the manufacturer's instructions. Absorbance was read at 450 nm, and fluorescence at ex/em = 540/590. BCAA levels in plasma or Skm tissue were measured using the Branched Chain Amino Acid Assay Kit (Abcam). The initial rate of lactate production was measured as previously described (Formentini *et al*, 2012).

For FFA and free glycerol determination, tissues were homogenized in 1 ml of 2-isopropanol in a TissueLyser (Qiagen). Five microliters of Skm/WAT extracts or 5 μl of mouse blood serum was used for FFA quantification using the Glycerol Quantification Kit (Sigma-Aldrich) at $A_{540}$. Quantification of specific lipid species was assessed by lipidomics (see below). All results were adjusted for exact protein contents.

### FFA β-oxidation

Four days after differentiation, primary myotubes derived from wt or ATPIF1$_{H49K}$|$^{T/H}$ mice or transfected $C_2C_{12}$ cells were incubated in serum-free α-MEM containing [9,10-$^3$H(N)] palmitic acid (Perkin Elmer, 0.2 μCi, final concentration = 20 μmol/l). After incubation, 100 μl of the culture medium was placed over an ion-exchange resin, and the Poly-Prep Chromatography column was washed with water. Intact FFAs (charged state) were retained by the resin, whereas the oxidized portion passed freely (Formentini *et al*, 2017a). The oxidized portion was measured in a scintillation counter with Ultima Gold LLT scintillation fluid (Perkin Elmer). All results were adjusted for total cellular protein content.

### 14C(u)-leucine, 14C(u)-isoleucine, and 14C(u)-glucose uptake and catabolism

For the measurement of substrate uptake, lipid synthesis, and $CO_2$ production, myocytes were incubated in HBSS containing 0.3 mM L-leucine, 0.3 mM L-isoleucine, or 5 mM d-glucose + 2 μCi/ml of labeled 14C(u)-L-leucine, 14C(u)-L-isoleucine, or 14C(u)-D-glucose (Perkin Elmer).

Uptake was measured as radioactivity incorporated in myocytes after 0, 1, 2, 3, 5, or 10 min. Lipid fractions were separated by a standard CHCl$_3$/MeOH extraction at 0, 2, and 4 h. For $CO_2$ production, incubation was carried out in flasks in the presence of a paper filter imbibed in 0.2 M KOH/NaOH solution. The C$^{14}$-uptake and C$^{14}$-incorporation into lipids or $CO_2$ were measured in a scintillation counter with Ultima Gold LLT scintillation fluid (Perkin Elmer).

### Quantitative lipidomics

Lipidomics was performed by Lipotype (https://www.lipotype.com/), Max Planck Institute of Molecular Cell Biology and Genetics in Dresden, Germany. Details on methods are detailed in the Expanded View Section.

### Real-time PCR

We measured the Skm expression levels of: osteonectin (SPARC); musclin (OSTN); LIF; fractalkine (FKN/CX3CL1); follistatin-like protein (FSTL1); TNFα; IL15; IL6; PPARδ; irisin (FNDC5); GLUT1; GLUT3 and PGC1α. RNA was purified using 100 mg of hindlimb muscles from 3 ATPIF1$_{H49K}$ and 3 wt mice following standard methods detailed in Expanded View Methods.

### 1D- and 2D-PAGE

Skm, WAT, brain and liver samples were freeze-clamped in liquid nitrogen. Tissue proteins were extracted in a buffer containing 50 mM Tris–HCl, pH 8.0, 1% NaCl, 1% Triton X-100, 1 mM DTT, 0.1% SDS, and 0.4 mM EDTA, supplemented with protease and phosphatase inhibitor cocktails. Lysates were clarified by centrifugation at 13,000 *g* for 15 min at 4°C. The resulting supernatants were fractionated by SDS–PAGE and transferred onto PVDF or

nitrocellulose membranes for immunoblot analysis. Blocking was performed with 5% nonfat dried milk in Tris-buffered saline with 1% Tween 20 (TBST) or TBST supplemented with 5% BSA. The primary monoclonal antibodies developed in our laboratory and used in this study were anti-β-F1-ATPase (1:20,000), anti-HSP60 (1:10,000), and anti-GAPDH (1:20,000) (Formentini *et al*, 2014). The antibodies specifically recognizing the human and mouse ATPIF1 proteins were used at a 1:200 dilution (Formentini *et al*, 2017b). Other antibodies used are listed in Table EV1. Blots were revealed using the Novex® ECL HRP Chemiluminescent reagent, and the intensity of the bands was quantified using a Bio-Rad GS-900 densitometer and ImageJ 1.51v analysis software.

Isoelectric focusing (IEF) was performed with 13-cm Immobiline DryStrips of 3–10 NL [not linear] pH gradient using an Ettan IPGPhor3 IEF unit (GE Healthcare). In brief, 200 μg of fresh-frozen Skm protein diluted in 250 μl of rehydration buffer (DeStreak Rehydration Solution, GE Healthcare) containing 0.5% of the corresponding IPG buffer (GE Healthcare) was loaded on the 13-cm strips. The equilibrated strips were transferred to the top of a 9% SDS–polyacrylamide gel. Electrophoresis was carried out using a Protean II XI system (Bio-Rad) with constant current (30 mA/gel) at 4°C for 3 h. Western blot analysis of the fractionated proteins was performed as described above.

### Immunoprecipitation assays

Respiratory Complex II subunit SDHA and acetylated proteins were immunocaptured from isolated mitochondria of Skm solubilized with 1% n-dodecyl-β-D-maltoside (DDM). Protein from cell lysates (400 mg) was incubated with 12 μg of the indicated antibody (SDHA or acetyl-k) bound to EZ View Red Protein G Affinity Gel (Sigma-Aldrich) at 4°C overnight. The beads were washed twice before proteins were eluted and fractionated on SDS–PAGE.

### Blue-native (BN) and clear-native (CN) PAGE

For BN PAGE, Skm isolated mitochondria from wt and ATPIF1$_{H49K}|^{T/H}$ mice were suspended in 50 mM Tris–HCl, pH 7.0, containing 1 M 6-aminohexanoic acid at a final concentration of 10 mg/ml. The membranes were solubilized by the addition of 10% digitonin (4:1 digitonin/mitochondrial protein). Next, 5% Serva Blue G dye in 1 M 6-aminohexanoic acid was added to the solubilized membranes. In the CN-loading buffer, the Serva Blue G dye was replaced by 0.1% Ponceau Red and 5.5% glycerol. In both BN and CN Native PAGE™, Novex® 3–12% Bis-Tris Protein Gels were loaded with 70 μg of mitochondrial protein. The electrophoresis was performed at a constant voltage of 70 V for 15 min, followed by 1 h at a constant amperage of 10 mA. BN cathode buffer: Tricine 50 mM, Bis-Tris 15 mM, pH = 7.0, Serva blue G 0.02%; BN anode buffer: Bis-Tris 50 mM, pH 7.0; CN-cathode buffer: 50 mM tricine, 15 mM Bis-Tris, 0.05% sodium deoxycholate, pH 7.0, CN-anode buffer: 50 mM Bis-Tris, pH 7.0.

### In-gel mitochondrial ETC complex activities

CN PAGE gels containing solubilized mitochondria from wt and ATPIF1$_{H49K}|^{T/H}$ mice were incubated with the following solutions for assessing the specific ETC complex activities: Complex I: 5 mM Tris–HCl, pH 7.4, 1 NTB tablet (5 mg; NitroBlue Tetrazolium Tablet), and 10 mg/ml NADH; Complex II: 5 mM Tris–HCl, pH 7.4, 1 NTB tablet (5 mg), 10 mM sodium succinate, and 250 mM phenazine methosulfate; Complex III: 50 mM phosphate buffer ($NaH_2PO_4$ + $NaHPO_4$, pH = 7.2) and 5 mg DAB; Complex IV: 50 mM phosphate buffer ($NaH_2PO_4$ + $NaHPO_4$, pH = 7.2), 5 mg DAB, and 5 mM reduced cytochrome C.

### Quantitative proteomics (iTRAQ)

Isobaric tag for relative and absolute quantitation (iTRAQ) analysis was carried out in the CBMSO Protein Chemistry Facility (ProteoRed, PRB3-ISCIII and UAM University, Spain), following standard protocols detailed in the Expanded View Section.

### Electron microscopy

Sample preparation was performed by the Electron Microscopy Facility at the CBMSO, UAM University, Spain. Skm tissue was fixed with 4% paraformaldehyde and 2% glutaraldehyde in 0.1 M phosphate buffer. It was then treated with 1% osmium tetroxide in water at 4°C for 1 h, dehydrated with ethanol, and embedded in TAAB 812 epoxy resin. Ultrathin 80-nm sections of the embedded tissue were obtained using an ultramicrotome Ultracut E (Leica) and mounted on carbon-coated copper 75-mesh grids. The sections were stained with uranyl acetate and lead citrate and examined at 80 kV in a JEOL JEM 1010 electron microscope. Images were recorded with a TemCam F416 (4k × 4K) digital camera from TVIPS.

### Immunofluorescence, confocal and optic microscopy

Hindlimb Skm from wt and ATPIF1$_{H49K}|^{T/H}$ mice was sliced, histologically prepared, and stained with hematoxylin/eosin by the Histology Facility at CNB-CSIC, UAM University, Spain. Deparaffination was performed at 60°C for 1 h, followed by hydration (xylene, EtOH 100%, EtOH 90%, EtOH 70% and distilled H$_2$O). C$_2$C$_{12}$ cells were fixed in 4% PFA. Blockage was performed with 3% goat serum in TBS 1× and 0.5% Triton X-100 at RT for 1 h. Dyes were incubated in 1% goat serum in TBS 1× and 0.5% Triton X-100. Stainings were as follows: ATPIF1 (1:500), β-F1-ATPase (1:10,000), BODIPY 493/503 (2 μM) for LD; DAPI (1:1,000) for nuclei; and Oil Red O (0.5%) for lipid staining. Complex I activity in slices was performed as indicated for the CN *in-gel* activity. Images were acquired on a Leica DMRE light microscope or by confocal microscopy using a Bio-Rad Radiance 2000 Zeiss Axiovert S100TV. ImageJ 1.51v software was used for quantification and image analysis.

### Statistical analyses

Statistical analyses were performed using a two-tailed Student's *t*-test. ANOVA and the Tukey's *post hoc* test were used for multiple comparisons, employing SPSS 17.0 and GraphPad Prism7 software packages. Bonferroni correction was applied to avoid multiple comparison errors. The results shown are the means ± SEM. $P < 0.05$ was considered statistically significant. The *n* used in each statistical test is indicated in the figure legends.

After normalization and filtering steps, proteomic and lipidomic data were analyzed by Gene Set Enrichment Analysis and visualized by heat map and enrichment map using GSEA v3.0 and Cytoscape v3.6.1 free software. For details on the GSEA parameter usage, see the GSEA web site (http://www.gsea-msigdb.org/gsea/index.jsp). For bioinformatic studies, the PyMOL Molecular Graphics System, Version 2.1.0 Schrödinger, LLC was used. Real-time PCR analysis was assessed using 7500 Real-Time PCR SDS 2.4 software.

## Assessment of mouse motor function

### Grip force/strength test

The grip force test was used to measure the maximum strength that could be performed by a mouse with its forelimbs by taking advantage of the animal's tendency to grasp to surfaces. One mouse at a time was left to grasp the metallic bars and then was gently pulled away until its grasp is broken. The pulling was performed at constant speed and sufficiently slow to permit the mouse to build up a resistance against it. The test was repeated five times per mouse, with at least 1 min elapsing between each of the five determinations per animal.

### Rotarod test

The rotarod test was performed in 3 days. Days 1 and 2 the mice were trained, and the third day the test was performed. In day 1, training consisted in 15 min at 13 rpms, and they were replaced in the rod if they fell off. After 30 min of test ,mice were replaced on the rod at run speeds ramping from 13 to 20 rpms for 15 min. In day 2, training consisted in run speed from 13 to 20 rpms for 15 min and the second run was ramping from 13 to 20 rpms in 180 s for 15 min. The test involved run speed ramping from 13 to 20 rpms in 180 s and then let the animals in the rod until they fell off. Their latency was measured. Mice that fell off four times within 60 s were discarded from the experiment.

### Open field test

Mice were left free to explore an open field arena (40 × 40 cm) for 10 min, and activity was recorded for analysis with ANY-maze software. To further explore fine alterations several aspects of the motor activity were scored: speed (mean and maximum), time mobile (expressed as percentage of the total time of the trial). Layout: open field area was virtually dissected in the following areas: center (inner square), outer region (everything but the center square), and corners (four triangular regions in the corners). Activity measurements were evaluated in each region as well as the entire arena. Abnormal phenotypes including thigmotaxis and anxiety-like behavior are studied by virtual dissection of the arena with focus on the corners and inner zone.

## Real-time PCR

RNA was purified using 100 mg of hindlimb muscles from 3 ATPIF1$_{H49K}$ and 3 wt mice following a standard TRIzol/chloroform method. Purified RNA (1 μg) was retrotranscribed into cDNA with the High-Capacity cDNA Reverse Transcription Kit. Real-time PCR was performed using the Fast SYBR Master Mix and ABI Prism 7900HT sequence detection system at the Genomics and Massive Sequencing Facility (CBMSO–UAM). The primers used to amplify the target genes are detailed in Table EV2. Actin and GAPDH were selected as housekeeping genes to normalize the mRNA levels. Standard curves with serial dilutions of pooled cDNA were used to assess the amplification efficiency of the primers and to establish the dynamic range of cDNA concentration for amplification. SDS 2.4 software was used for data collection, and the relative expression of the mRNAs was determined with the comparative $\Delta\Delta C_t$ method.

## Quantitative lipidomics

### Lipid extraction for MS lipidomics

Mass spectrometry-based lipid analysis was performed by Lipotype GmbH as described (Sampaio et al, 2011). Samples were spiked with internal lipid standard mixture containing the following: cardiolipin 16:1/15:0/15:0/15:0 (CL), ceramide 18:1;2/17:0 (Cer), diacylglycerol 17:0/17:0 (DAG), hexosylceramide 18:1;2/12:0 (HexCer), lyso-phosphatidate 17:0 (LPA), lyso-phosphatidylcholine 12:0 (LPC), lyso-phosphatidylethanolamine 17:1 (LPE), lyso-phosphatidylglycerol 17:1 (LPG), lyso-phosphatidylinositol 17:1 (LPI), lyso-phosphatidylserine 17:1 (LPS), phosphatidate 17:0/17:0 (PA), phosphatidylcholine 17:0/17:0 (PC), phosphatidylethanolamine 17:0/17:0 (PE), phosphatidylglycerol 17:0/17:0 (PG), phosphatidylinositol 16:0/16:0 (PI), phosphatidylserine 17:0/17:0 (PS), cholesterol ester 20:0 (CE), sphingomyelin 18:1;2/12:0;0 (SM), and triacylglycerol 17:0/17:0/17:0 (TAG). After CHCl3/MeOH extraction, the organic phase was transferred to an infusion plate and dried in a speed vacuum concentrator. The first step dry extract was resuspended in 7.5 mM ammonium acetate in chloroform/methanol/propanol (1:2:4, V:V:V) and the second step dry extract in a 33% ethanol solution of methylamine in chloroform/methanol (0.003:5:1; V:V:V). All liquid handling steps were performed using the Hamilton Robotics STARlet robotic platform with the Anti Droplet Control feature for organic solvent pipetting.

### MS data acquisition

Samples were analyzed by direct infusion on a Q Exactive mass spectrometer (Thermo Scientific) equipped with a TriVersa NanoMate ion source (Advion Biosciences). Samples were analyzed in both positive and negative ion modes with a resolution of Rm/z = 200 = 280,000 for MS and Rm/z = 200 = 17,500 for MS/MS experiments in a single acquisition. MS/MS was triggered by an inclusion list encompassing corresponding MS mass ranges scanned in 1-Da increments. Both MS and MS/MS data were combined to monitor CE, DAG, and TAG ions as ammonium adducts; PC, PC and O-, as acetate adducts; and CL, PA, PE, PE O-, PG, PI, and PS as deprotonated anions. MS only was used to monitor LPA, LPE, LPE O-, LPI, and LPS as deprotonated anions; Cer, HexCer, SM, LPC and LPC O- were used as acetate adducts.

## Quantitative proteomics (iTRAQ)

### Protein digestion

In solution digestion: After denaturation of protein with 8 M urea, samples were reduced and alkylated with 10 mM DTT (1 h at 37°C) and 50 mM iodoacetamide (1 h at room temperature), respectively. Next, samples were diluted to reduce the urea concentration below

1.4 M and digested using sequencing grade trypsin (Promega) overnight at 37°C using a 1:5 (w/w) trypsin/protein ratio. Whole supernatants were dried down and then desalted on OASIS C18 columns (Waters) until the mass spectrometric analysis.

### iTraq labeling and high-pH fractionation

The resultant peptide mixture from desalted protein tryptic digests (100 μg) was dissolved in 30 μl of 0.5 M triethylammonium bicarbonate (TEAB), pH 8, and labeled using the iTRAQ reagent 4plex Multi-plex kit (Applied Biosystems). Labeling was stopped by the addition of 0.1% formic acid. The obtained "4plex-labeled mixture" was analyzed by RP-LC-MS/MS to check the efficiency of the labeling. The sample was then fractionated using the high-pH, reversed-phase peptide fractionation kit (Pierce, Thermo Scientific). The sample was re-swollen in 0.1% TFA and then loaded onto an equilibrated, high-pH, reversed-phase fractionation spin column. A step gradient of increasing acetonitrile concentrations in a volatile high-pH solution was applied to the columns to elute bound peptides into nine different fractions (5–80% acetonitrile) collected by centrifugation. The fractions obtained were dried and stored until analysis by mass spectrometry for quantification.

### Quantitative analysis by RP-LC-MS/MS

The fractions were resuspended in 10 μl of 0.1% formic acid and analyzed by RP-LC-MS/MS in an Easy-nLC II system coupled to an ion trap LTQ-Orbitrap-Velos-Pro hybrid mass spectrometer (Thermo Scientific, Waltham, Massachusetts, Estados Unidos). Peptides were concentrated (on-line) by reverse phase chromatography using a 0.1 × 20 mm C18 RP precolumn (Proxeon) and then separated using a 0.075 × 250 mm C18 RP column (Proxeon) operating at 0.3 μl/min. Peptides were eluted using a 90-min dual gradient from 5 to 25% solvent B in 68 min, followed by a gradient from 25 to 40% solvent B over 90 min (solvent A: 0.1% formic acid in water, solvent B: 0.1% formic acid, 80% acetonitrile in water). ESI ionization was performed using a Nano-bore emitters Stainless Steel ID 30 μm (Proxeon) interface. The instrument method consisted of a data-dependent top-20 experiment with an Orbitrap MS1 scan at a resolution ($m/\Delta m$) of 30,000, followed by 20 high energy collision dissociation (HCD) MS/MS mass analysis in an Orbitrap at 7,500 ($\Delta m/m$) resolution. MS2 experiments were performed using HCD to generate high resolution and high mass accuracy MS2 spectra. The minimum MS signal for triggering MS/MS was set to 500. The lock mass option was enabled for both MS and MS/MS mode, and the polydimethylcyclosiloxane ions (protonated $(Si(CH_3)_2O))_6$; $m/z$ 445.120025) were used for internal recalibration of the mass spectra. Peptides were detected in survey scans from 400 to 1,600 amu (1 μscan) using an isolation width of 2 u (in mass-to-charge ratio units), normalized collision energy of 40% for HCD fragmentation, and dynamic exclusion applied during 30-s periods. Precursors of unknown or +1 charge state were rejected.

### Proteomic and lipidomic data analyses

### Proteomics

Peptide identification from raw data was carried out using a PEAKS Studio X search engine (Bioinformatics Solutions Inc.). A database search was performed against uniprot-Mus musculus.fasta (decoy-fusion database). The following constraints were used for the searches: tryptic cleavage after Arg and Lys, up to two missed cleavage sites, and tolerances of 20 ppm for precursor ions and 0.05 Da for MS/MS fragment ions, and the searches were performed allowing optional Met oxidation, Cys carbamidomethylation, and iTRAQ reagent labeling at the N-terminus and lysine residues. False discovery rates (FDRs) for peptide spectrum matches (PSMs) were limited to 0.01. Only those proteins with at least two distinct peptides being discovered from LC/MS/MS analyses were considered reliably identified and sent to be quantified. Quantitation of iTRAQ-labeled peptides was performed with a PEAKS Studio X search engine. The −10LgP, quality and reporter ion intensity were used for spectrum filter and significance (ANOVA method) for protein filter. For protein quantification, we considered protein groups for peptide uniqueness and used only unique peptides, and the modified peptides were excluded.

### Lipidomics

Data were analyzed with lipid identification software LipidXplorer (Lipotype). Only lipid identifications with a signal-to-noise ratio > 5 and a signal intensity 5-fold higher than that in corresponding blank samples were considered for further data analysis.

## Data availability

Skm and WAT iTRAQ proteomic data are available via ProteomeXchange with identifier PRIDE PXD017621; http://www.ebi.ac.uk/pride/archive/projects/PXD017621 (Skm), PXD017683; http://www.ebi.ac.uk/pride/archive/projects/PXD017683 (Skm + HFD) and PXD017678; http://www.ebi.ac.uk/pride/archive/projects/PXD017678 (WAT). Skm and WAT lipidomic data are available via Figshare: https://doi.org/10.6084/m9.figshare.11872077.v1

**Expanded View** for this article is available online.

## Acknowledgements

We thank The CBMSO Protein Chemistry, Electron Microscopy, Flow Cytometry, Confocal Microscopy, Genomic and Animal House Facilities. We acknowledge Marta Gómez de Cedrón Cardeñosa from IMDEA Food; Professor T. Ciaraldi from VA Medical Center, UCSD; Dr. Javier García-Bermúdez from Rockefeller University; and Dr. David Abia from the Bioinformatics Unit of the CBMSO (CSIC/UAM) for advice. This work was supported by grants from Ministerio de Economía, Industria y Competitividad, MINECO, Spain (SAF2016-76028-R and SAF2016-75916-R) and Centro de Investigación Biomédica en Red (CIBER) de Enfermedades Raras, Spain (CB06/07/0017). LF is supported by the Ramón y Cajal Spanish programme (RYC-2013-13693). CSG and JHM are supported by MINECO (BES-2017-079909) and Comunidad de Madrid (CAM30534) fellowships, respectively. The CBMSO Protein Chemistry Facility was supported by PT17/0019 grant.

## Author contributions

LF designed the study; conceived, performed and analyzed experiments; wrote the manuscript; and secured funding. CS-G performed the experiments, contributed to the analysis and reviewed/edited the manuscript. CN-T, JCHM and MPP researched the data and critically revised the article. SSS researched

the data. ARM reviewed/edited the manuscript. JMC contributed to the conception and final revision of the manuscript and to secure funding. All the authors approved the final version of the manuscript.

## Conflict of interest

The authors declare that they have no conflict of interest.

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
