## [Review Process File · The EMBO Journal]

Dysfunctional oxidative phosphorylation shunts branched-chain amino acid catabolism onto lipogenesis in skeletal muscle

Cristina Sanchez-Gonzalez, Cristina Nuevo-Tapioles, Juan Cruz Herrero Martin, Marta Pereira, Sandra Serrano Sanz, Ana Ramirez de Molina, Jose Cuezva, and Laura Formentini
DOI: [10.15252/embj.2019103812](https://doi.org/10.15252/embj.2019103812)

Review Timeline:

Submission Date:	23rd Oct 19
Editorial Decision:	20th Nov 19
Revision Received:	3rd Mar 20
Editorial Decision:	27th Mar 20
Revision Received:	8th Apr 20
Accepted:	27th Apr 20

Editor: Elisabetta Argenzio

Transaction Report:

Thank you for submitting your manuscript entitled "Dysfunctional muscle oxidative phosphorylation shunts branched-chain amino acid catabolism into lipogenesis" [EMBOJ-2019-103812] to The EMBO Journal. Your study has been sent to three referees for evaluation, whose reviews are enclosed below.

As you can see, the referees find your work the potentially interesting and appreciate the new mouse model presented in the study. However, they also raise several critical points that have to be addressed before they can support the publication of your work in The EMBO Journal. In particular, they need you to further investigate the role of branched chain amino acids in lipid-droplet formation in your model system. Furthermore, reviewer #1 asks you to functionally validate the enzymes identified in the proteomic screen and to further investigate how mitochondrial respiratory complexes I and II contribute to ROS generation; referee #2 requests you to assess the motor functions of ATP1F1H49K mice and to measure food intake and activity in edaravone-treated animals.

Given the overall interest of your study, I would like to invite you to revise the manuscript in response to the referee reports. I should also note that conclusively addressing these all the referees' points is essential for publication in The EMBO Journal.

I realize that addressing all criticisms may take a long time and be technically challenging. I would therefore understand if you would choose not to undergo an extensive revision here and rather pursue a submission at an alternative venue, in which case please inform us at your earliest convenience.

When preparing your letter of response to the referees' comments, bear in mind that this will form part of the Review Process File and will be available online to the community. For more details on our Transparent Editorial Process, please visit our website:

http://emboj.embopress.org/about#Transparent_Process.

We generally grant three months as standard revision time. As a matter of policy, competing manuscripts published during this period will not negatively impact on our assessment of the conceptual advance presented by your study. Nevertheless, please contact me as soon as possible upon publication of any related work.

Referee #1:

Sánchez-González et al. investigate functional consequences of defective respiration in muscle mitochondria on muscle and whole-body metabolism. The authors generate a mouse model that expresses an activated form of mitochondrial ATP synthase inhibitor ATP1F1 selectively in skeletal muscle. Using diverse approaches including proteomics, lipidomics, mitochondrial activity and whole-body metabolism assays, the authors identify widespread metabolic alterations. In particular, the lipid content of muscle increases accompanied by obesity and a diabetic phenotype. Increased lipogenesis is linked to increased BCAA catabolism. Analyzing activity and structure of different mitochondrial complexes, the authors suggest that increased ROS production also contributes to these metabolic alterations. Finally, a clinically approved drug, edaravone, which has antioxidant activity, is found to reverse the alterations caused by ATP synthase inhibition, which points to its potential use in treating metabolic diseases.

Understanding metabolic consequences of dysregulated mitochondrial function is of great interest, given the clear implication of mitochondrial dysfunction to multiple human diseases. This manuscript is an interesting and relevant study of how consequences of perturbed OXPHOS in Skm. The authors use diverse approaches and present a large amount of data to investigate these phenotypes, ranging from cellular, organ and organismal metabolism to proof-of-principle experiments for a potential new treatment of metabolic disorders. However, the manuscript also leaves the impression that the authors try to cover too many angles, which leaves several claims correlative or unsubstantiated. Therefore, several conclusions require further experimental support. Because the manuscript is rather complex, data peripheral to the central message could alternatively be removed or their correlative nature at least be discussed in a more balanced way.

Major points:

1. The paper's title 'dysfunctional muscle oxidative phosphorylation shunts branched-chain amino acid catabolism into lipogenesis' is not supported by the data. In fact, I am not sure whether the title reflects the central message of the paper. Expression of ATP1F1 does not change levels of the rate-limiting enzyme of BCAA catabolic pathways, BCKDH (Fig. 1K, EV2). It is therefore questionable whether changes in the levels of downstream enzymes increase pathway flux. No functional validation of the changes in enzyme levels observed in the proteomics data are provided. Instead, the authors address BCAA catabolism first by quantifying formation of lipid droplets in cells fed palmitate +/- AA. This experiment does not address whether cells catabolize BCAA. The increase in lipid-droplet formation could e.g. be because amino acid starvation influences synthesis of fat from palmitate. Thus, the experiment should be performed by specifically adding / removing BCAA, alone or together with palmitate. The authors further quantify incorporation of carbon from ¹⁴C-labelled AA into lipids. According to the methods part, BCAA are not labeled specifically, which again precludes any conclusions about BCAA or other AA. The results could be explained by increased production of glycerol from AA as a backbone for glycerolipid synthesis.

2. I am confused about the experiments addressing mitochondrial ROS production. The authors demonstrate in ATP1F1-expressing mice increased levels of CII without an accompanying change in activity, suggesting decreased function of individual CII complexes. The authors nicely link this to increased acetylation of SDHA. By contrast, activity and levels of CI are not altered (Fig. 1E, 4E, EV4B - D). However, the authors observe a subassembly of CI, which is suggested to represent an early stage of ROS-mediated CI degradation that might correlate with alterations in cardiolipin levels. The authors neither explain whether changes in CII activity is causal for ROS, nor why CI, although not changing in activity or abundance, should contribute to ROS generation. The cardiolipin data are entirely correlative.

3. In Fig. 7, most experiments lack control groups for the edaravone-treated animals.

Minor points:

1. Do ATP levels change in ATP1F1-expressing Skm? Does moderate pharmacological inhibition of CV recapitulate phenotypes of ATP1F1 expression (e.g. increased lipid droplet formation)?

2. Fig. 1D 'expression of mouse-ATP1F1 in Skm was null' - change to 'not detectable'

3. Fig 2A: Regarding the obese phenotype, does a decrease in Skm mitochondrial OXPHOS alter activity / energy expenditure of the animals?

4. Fig 2I: ATPase inhibition causes a slight increase in PDH phosphorylation, which leads the authors to pursue BCAA instead of glucose as the carbon source for lipogenesis. More direct evidence, e.g. decreased glucose consumption, is required to support this.

5. Fig. 3: The lipidomics data for individual lipid species should be included in the supplement.

6. Fig. 3J: Increased TNF α levels are an interesting but correlative observation. The authors should provide experimental evidence that because of increased TNF α 'as a consequence, Skm-specific impairment of OXPHOS also altered de novo lipid synthesis in WAT' or remove statement.

7. Fig. 3N-P: Along the same lines, 'highlighting muscle mitochondrial activity as an organ-to-organ regulator of lipogenesis' is not supported by the data. The effects on WAT could also be due to whole-body metabolic alterations.

8. It is interesting, that inhibition of CV in several ways has opposite effects to metformin, whose effect on metabolism is suggested to be in part due to inhibition of CI. Maybe the authors could discuss this?

9. The readability of the manuscript could be improved: Many experiments and results are mentioned superficially and should be explained in more detail. It would also help if the arrangement of figure panels followed a more logical order.

Referee #2:

ristina Sanchez-Gonzalez and colleagues demonstrate a new skeletal muscle specific mouse model using a conditional human ATP1F1 active mutant system similar to what they have previously

published in EMBO in other tissue settings. This mouse model is very interesting since it provides a genetic approach for the tissue specific inhibition of mitochondrial ATP production that helps to untangle the complicated inter-organ metabolic relationships that occur during the onset of insulin resistance and obesity. Overall this work is very interesting and is extensively characterized at the molecular level using multiple in vitro, in vivo, and ex vivo approaches. They also identify a drug, edaravone, an FDA approved drug in ALS that is thought to be an antioxidant modifier of ROS, which is capable of preventing the onset of some aspects of dysfunction demonstrated in their mouse model. This is intriguing data and adds to the literature on this drug as a potential metabolic therapy that has already been under investigation as a possible modifier of diabetic nephropathy and ischemia reperfusion injury. This manuscript also adds interesting data to the literature on the role of BCAA and lipid metabolism in insulin resistance and obesity.

Major Comments

1) There is no assessment of the motor function, are there functional deficits in locomotion, running, strength? It would be interesting to run these mice through some aspects of physiological motor behavior in the absence of HFD treatment to see if at baseline these mice have any physiological impairment of skeletal muscle. Possible examples include grip strength, metabolic cage/activity tracking, open field behavior, running wheel spontaneous exercise measurements, or treadmill forced maximal running testing.

2) I am not convinced by the argument that BCAAs are the main acetyl coA source for lipid droplet formation by the in vitro C2C12 experiments by comparison to pyruvate or citrate sources. It seems that this claim is primarily based on a western blot showing increased phosphorylation of PDH, however this does not demonstrate a complete block of the pathway, and the lack of a difference in lipid droplet formation upon growth in an AA-free medium. However limiting all amino-acids will suppress anabolic metabolism. This experiment is not isolating the effects of BCAA metabolism specifically. It may be interesting to repeat the carbon labeling experiments with labeled glucose or glutamine which may also demonstrate significantly more enrichment of C14 lipids in the setting of ATP1F1 overexpression in C2C12 cells.

3) Is it possible that the effect of edaravone is simply delaying the onset of insulin resistance in these models? Looking at the curves of the ITT and GTT, it looks like both the WT and the transgenic mice are fairly insulin sensitive upon edaravone treatment at 60 days. Have you measured food intake and activity to see if edaravone is changing either of these parameters to impact the onset of HFD-induced disease in both groups?

4) These data suggest that inhibition of mitochondrial ATP production in skeletal muscle is sufficient to increase susceptibility to whole body lipid metabolic changes associated with insulin resistance. It does not, however tackle the necessity of this change, it may be that there are multiple metabolic tissues for which a similar change will result in a similar metabolic phenotype.

Minor points:

In the abstract, branched-chain amino acids is abbreviated as BCCA instead of BCAA.

Transition to lipodystrophy in the second paragraph of the introduction - not sure this is the right term, typically lipodystrophy is reserved for the rare genetic lipodystrophies and isn't really applied to obesity. Why not stay with obesity, or metabolic syndrome? Similarly in the first sentence on page 9 -"ATP-synthase dependent dyslipidemia in muscle" I think a different wording is required since dyslipidemia implies the blood lipid profile.

In the materials section, the statement Mitoq was a "gentle concession" should probably be amended to a gift.

Figure 1 A there is an n= that doesn't appear to have anything following it.

Figure 1 D: is there an increase in liver and WAT mouse endogenous ATPIF by densitometry analysis? If so why do you think that would that be?

Is the first section of Figure 6 E real data or also a schematic like in A? If it is data, are there time points that could be highlighted on the graph to make that distinction more pronounced? Also this data is shown in the bar graph so I am not sure it is necessary.

Referee #3:

The manuscript entitled "Dysfunctional muscle oxidative phosphorylation shunts branched-chain amino acid catabolism into lipogenesis" by Sánchez-González, et al. provides a deep exploration of metabolic consequences of ATP synthase inhibition. The authors generate an interesting mouse model expressing an active form of ATPIF1 in striatal muscle. The mice present with increased weight gain, including increased levels of visceral fat, and develop insulin resistance faster. Muscle fatty acid oxidation decreases while BCAA catabolism increases leading to acetyl-CoA accumulation and reduced OXPHOS. Complementary alterations are observed in adipose tissue and supported by proteomics and metabolic data, including lipidomic analysis. Finally, a screen identifies edaravone as a driver of increased FA oxidation and OCR which prevents diet-induced obesity.

The study is fairly detailed and well-executed. At times the paper reads long, but the authors are covering a lot of metabolic ground. Generally, the respiration and mitochondrial studies are stronger than the lipogenesis data, but the study highlights an interesting contrast between muscle and adipose BCAA catabolism. I have some conceptual questions as well as some technical issues that should be addressed.

1. Why does increased mitochondrial AcCoA (which is generated by BCAA catabolism) increase tissue AcCoA? For that matter, why does AcCoA increase lipogenesis. FASN and other enzymes are increased, but is this due to acetylation? Inflammation? Please clarify in the manuscript.
2. Quantifying BCAA uptake in the ATPIF1H49K cells would be very informative. Is BCAA uptake higher in ATPIF1H49K cells? How does FASN inhibition or culture in delipidated media affect BCAA catabolism?
3. The authors attempt to show that BCAAs are essential for LD formation by culturing cells in AA-free media. Is this simply mT or inhibition/autophagy-driven rather than BCAA-specific?
4. Visceral AT is lower with edaravone treatment. Is muscle whitening decreased? Does this treatment decrease skeletal muscle BCAA abundance?

Minor:

1. Fig 7C should be PGC1 α .
2. Fig 6E - error bars should be present?
3. Figure layout (letter panels) is tough to follow and should be organized better, though I understand there is a lot of data.
4. Figure 6C poorly depicts the tracing study. Why 3 circles if only 2 isotopes present per molecule?

Referee #1:

Sánchez-González et al. investigate functional consequences of defective respiration in muscle mitochondria on muscle and whole-body metabolism. The authors generate a mouse model that expresses an activated form of mitochondrial ATP synthase inhibitor ATP1F1 selectively in skeletal muscle. Using diverse approaches including proteomics, lipidomics, mitochondrial activity and whole-body metabolism assays, the authors identify widespread metabolic alterations. In particular, the lipid content of muscle increases accompanied by obesity and a diabetic phenotype. Increased lipogenesis is linked to increased BCAA catabolism. Analyzing activity and structure of different mitochondrial complexes, the authors suggest that increased ROS production also contributes to these metabolic alterations. Finally, a clinically approved drug, edaravone, which has antioxidant activity, is found to reverse the alterations caused by ATP synthase inhibition, which points to its potential use in treating metabolic diseases.

Understanding metabolic consequences of dysregulated mitochondrial function is of great interest, given the clear implication of mitochondrial dysfunction to multiple human diseases. This manuscript is an interesting and relevant study of how consequences of perturbed OXPHOS in Skm. The authors use diverse approaches and present a large amount of data to investigate these phenotypes, ranging from cellular, organ and organismal metabolism to proof-of-principle experiments for a potential new treatment of metabolic disorders. However, the manuscript also leaves the impression that the authors try to cover too many angles, which leaves several claims correlative or unsubstantiated. Therefore, several conclusions require further experimental support. Because the manuscript is rather complex, data peripheral to the central message could alternatively be removed or their correlative nature at least be discussed in a more balanced way.

We thank the Reviewer for her/his positive feedback and for recognizing the relevance of understanding the role of dysfunctional mitochondria in human diseases. We apologize for the lack of clarity that she/he highlighted. Many points are now improved for the message to be clear and accurate.

Major points:

Q1. The paper's title 'dysfunctional muscle oxidative phosphorylation shunts branched-chain amino acid catabolism into lipogenesis' is not supported by the data. In fact, I am not sure whether the title reflects the central message of the paper. Expression of ATP1F1 does not change levels of the rate-limiting enzyme of BCAA catabolic pathways, BCKDH (Fig. 1K, EV2). It is therefore questionable whether changes in the levels of downstream enzymes increase pathway flux. No functional validation of the changes in enzyme levels observed in the proteomics data are provided. Instead, the authors address BCAA catabolism first by quantifying formation of lipid droplets in cells fed palmitate +/- AA. This experiment does not address whether cells catabolize BCAA. The increase in lipid-droplet formation could e.g. be because amino acid starvation influences synthesis of fat from palmitate. Thus, the experiment should be performed by specifically adding / removing BCAA, alone or together with palmitate. The authors further quantify incorporation of carbon from ¹⁴C-labelled AA into lipids. According to the methods part, BCAA are not labeled specifically, which again precludes any conclusions about BCAA or other AA. The results could be explained by increased production of glycerol from AA as a backbone for glycerolipid synthesis.

A1. We appreciate the insightful and helpful comments of the Reviewer. We have now performed functional validations of increased BCAA pathway flux and better defined the relationship between BCAA catabolism and increased lipogenesis.

In order to demonstrate that lipid accumulation upon OXPHOS inhibition is directly related to augmented BCAA catabolism, we overexpressed the human protein ATP1F1_{H49K} in mouse myocytes grown in media with or without leucine, isoleucine and valine. Despite in complete media the inhibition of the ATP synthase during palmitate supplementation triggered a higher number of BODIPY-positive lipid droplets, upon BCAA deprivation no differences were observed between the two genotypes (see new Fig 2T and EV3E). Moreover, in line with our hypothesis, plasma levels of BCAA were augmented in Low_{OXPHOS} mice compared to wt (new Fig 3A), and the uptake of 14C(u)-leucine and 14(u)-isoleucine increased in myocytes expressing ATP1F1_{H49K} (new Fig 3B). Interestingly, no differences in the Skm BCAA amounts were detected between the two genotypes (new Fig 3C), suggesting an increase in BCAA muscular catabolism in Low_{OXPHOS} mice. Consistently, in this situation, 14C(u)-leucine and 14(u)-isoleucine oxidation to CO₂ was increased compared to control (see new Fig 3D). Moreover, when myocytes were administrated with 14C(u)-leucine and 14(u)-isoleucine and the lipid fraction extracted, a higher concentration of C14-lipids was observed upon OXPHOS inhibition (new Fig 3E).

We also took into account a possible role for a glycolytic shift in Low_{OXPHOS} mice, that might explain increased production of glycerol as a backbone for glycerolipid synthesis and increased levels of acetyl-CoA. As previously reported (Formentini et al., Mol Cell 2012), the inhibition of mitochondrial ATP production exerted by ATP1F1_{H49K} caused a rewiring of energy metabolism through an increased glycolysis (see new Fig 2J and EV3B), with the aim to maintain Skm ATP levels (new Fig EV3C). In this regards, Skm levels of free glycerol were increased in Low_{OXPHOS} mice compared to control (new EV3D), what certainly contributes to the lipogenic shift observed.

However, despite a slight increase in 14C(u)-glucose uptake (see new Fig 2K), ATP1F1_{H49K} expressing myocytes showed a reduced total oxidation of 14C(u)-glucose to CO₂ in comparison to controls (see new Fig 2L). This could be due to the role of acetyl-CoA as a metabolic sensor able to allosterically inactivate enzymes involved in its synthesis, such as the pyruvate dehydrogenase complex (PDH). Consistently, we found that PDH was phosphorylated in Low_{OXPHOS} mice, and pyruvate was rerouted to lactate production (new Fig 2N), thus ruling out glycolysis as the main source of acetyl-CoA during OXPHOS inhibition.

We thank the Reviewer for the substantial improvement that her/his suggestions have provided to the manuscript. We have now accommodated all these changes in Figure, Result and Discussion sections. (See also the answer to R1, major point 2).

Q2. I am confused about the experiments addressing mitochondrial ROS production. The authors demonstrate in ATP1F1-expressing mice increased levels of CII without an accompanying change in activity, suggesting decreased function of individual CII complexes. The authors nicely link this to increased acetylation of SDHA. By contrast, activity and levels of CI are not altered (Fig. 1E, 4E,

EV4B - D). However, the authors observe a subassembly of CI, which is suggested to represent an early stage of ROS-mediated CI degradation that might correlate with alterations in cardiolipin levels. The authors neither explain whether changes in CII activity is causal for ROS, nor why CI, although not changing in activity or abundance, should contribute to ROS generation. The cardiolipin data are entirely correlative.

A2. We apologize for the lack of clarity and thank the Reviewer for the interesting comment that helped to improve the mechanism of the ATP synthase-dependent increase in ROS production. We now provide demonstration that the observed burst in ROS is related with the acetylation and inhibition of SDHA. We treated ATP1F1_{H49K} expressing myocytes with specific ETC inhibitors (new Fig 5I). As expected, both rotenone and antimycin A, inhibiting CI and CIII respectively, increased ROS production (new Fig 5I). However, the treatment of the ATP1F1_{H49K} cultures with malonate, a known inhibitor of CII, reverted ROS amounts to the levels of wt (new Fig 5I), indicating a direct participation of CII in the generation of ROS upon ATP synthase inhibition. Malonate is known to inhibit CII at flavin site in SDHA. To understand if ROS from CII arose only from the flavin site, we repeated the experiment with carboxin, that inhibits CII at the ubiquinone-binding site. Interestingly, only malonate but not carboxin prevented the ATP1F1-dependent ROS production (new Fig 5J), suggesting a specific role for SDHA in this event. In this context, we now support the idea that CI is not causal for ROS production and that the early state of CI-disassembly is a consequence of the CII-dependent ROS.

As suggested by the Reviewer, we removed the cardiolipin data that appeared just correlative and introduced the new data in Fig 5I, J, and in Result and Discussion sections.

Q3. In Fig. 7, most experiments lack control groups for the edaravone-treated animals.

A3. We apologize for the inconvenient. New Fig 7 now accommodates data about the four conditions of the *in vivo* experiments: wt; LOW_{OXPHOS}; wt+edaravone and LOW_{OXPHOS} +edaravone mice.

Minor points:

mQ1. Do ATP levels change in ATP1F1-expressing Skm?

mA1. We measured the Skm levels of ATP and found no differences between wt and LOW_{OXPHOS} mice (new Fig EV3C). In this regard, we support the idea that the inhibition of mitochondrial activity exerted by ATP1F1_{H49K} caused a rewiring of energy metabolism through an increased glycolysis (see new Fig 2J, N and EV3B), with the aim to maintain Skm ATP levels (new Fig EV3C).

Does moderate pharmacological inhibition of CV recapitulate phenotypes of ATP1F1 expression (e.g. increased lipid droplet formation)?

Yes. Interestingly, the inhibition of the activity of the ATP synthase, both by the expression of its inhibitor ATP1F1_{H49K}, and pharmacological inhibition (5 μ M oligomycin), resulted in a higher number of BODIPY-positive lipid droplets in comparison to the control (new Fig 2T, 2U and EV3E, F). We thank the Reviewer to help us to demonstrate that the increase in lipid droplet upon palmitate supplementation is not due to a side effect of the ATP1F1_{H49K} expression but is a general trait of the CV inhibition.

mQ2. Fig. 1D 'expression of mouse-ATPIF1 in Skm was null' - change to 'not detectable'

mA2. We have now changed “null” to “not detectable”.

mQ3. Fig 2A: Regarding the obese phenotype, does a decrease in Skm mitochondrial OXPHOS alter activity / energy expenditure of the animals?

mA3. As requested also by Reviewer 2 (see the answer to R2, major point 1), we performed motor function and activity tests to unveil possible alterations due to the inhibition of the Skm OXPHOS. No significant alterations in food intake (new EV5B), Rotarod test (new Fig 3G) and Open Field test (new Fig 3H and EV4) were noticed between wt and LowOXPHOS animals; however, upon ATP synthase inhibition the mice displayed reduced performances in the Grip Force test after fatigue (Fig 3I), suggesting muscular weakness.

mQ4. Fig 2I: ATPase inhibition causes a slight increase in PDH phosphorylation, which leads the authors to pursue BCAA instead of glucose as the carbon source for lipogenesis. More direct evidence, e.g. decreased glucose consumption, is required to support this.

mA4. As described in answer to major point 1, we have now provided data of the rewiring of energy metabolism through an increased glycolysis upon mitochondrial inhibition (see new Fig 2J). This is also accompanied by a slight increase in 14C(u)-glucose uptake (new Fig 2K). However, and consistent with the phosphorylation and inhibition of the PDH, ATP1F1_{H49K} expressing myocytes showed a reduced total oxidation of 14C(u)-glucose to CO₂ in comparison to controls (see new Fig 2L). Consistently, we found that pyruvate was rerouted to lactate production (new Fig 2N), thus ruling out glycolysis as the main source of acetyl-CoA during OXPHOS inhibition.

mQ5. Fig. 3: The lipidomics data for individual lipid species should be included in the supplement.

mA5. We have now included the Skm and WAT lipidomic data for individual species as a supplementary table.

mQ6. Fig. 3J: Increased TNF α levels are an interesting but correlative observation. The authors should provide experimental evidence that because of increased TNF α 'as a consequence, Skm-specific impairment of OXPHOS also altered de novo lipid synthesis in WAT' or remove statement.

mA6. Following the suggestion of the Reviewer we have removed the statement that TNF α could be related to alterations in WAT. In line with previous findings (Formentini L. et al, *Diabetologia* 2017), the revised version of the manuscript accommodates the TNF α data as a possible contributor of the inflammatory environment that favors the setting of IR.

mQ7. Fig. 3N-P: Along the same lines, 'highlighting muscle mitochondrial activity as an organ-to-organ regulator of lipogenesis' is not supported by the data. The effects on WAT could also be due to whole-body metabolic alterations.

mA7. Following the recommendation of the Reviewer we have removed the statement and included a sentence in the Result section about the possibility that the observed changes in WAT could also be due to whole-body metabolic alterations.

mQ8. It is interesting, that inhibition of CI in several ways has opposite effects to metformin, whose effect on metabolism is suggested to be in part due to inhibition of CI. Maybe the authors could discuss this?

mA8. This is an interesting point, despite quite out of the scope of our paper. First of all, the mechanism of metformin action still remains controversial. The reported dose of metformin inhibiting CI is significantly higher than the one used for treating T2D. This could suggest that the metformin effect on reducing IR is more related to its effect on the AMPK/mTOR pathways rather than to a direct effect on mitochondrial activity. Moreover, also the effects on AMPK and mitochondria are still object of debate (*PLoS One*. 2014 Jun 20;9(6):e100525; *Cell Reports*. 2019 29, 1511–1523; *Front Endocrinol*. 2019 May 7;10:294; etc..). Some authors stressed that metformin alters mitochondrial dynamics and biogenesis, what may be prevalent in comparison to CI inhibition, thus resulting in improving mitochondrial activity. Moreover, it is not surprising that the inhibition of OXPHOS at different points may result in opposite effects. For instance, blocking CI by rotenone induces a drop in mitochondrial membrane potential while oligomycin induces a state of transient membrane hyperpolarization. Moreover, inhibiting CI is not the same that inhibiting complex V in terms of ATP production. When CI is inhibited, electrons can enter the ETC through CII thus resulting in ATP generation, what is not possible when the ATP synthase is inhibited.

mQ9. The readability of the manuscript could be improved: Many experiments and results are mentioned superficially and should be explained in more detail. It would also help if the arrangement of figure panels followed a more logical order.

mA9. We apologize for these errors. We have now re-arranged the figures, removed the correlative data and the manuscript has been revised to clarify the message and made it more focused.

Referee #2:

Cristina Sanchez-Gonzalez and colleagues demonstrate a new skeletal muscle specific mouse model using a conditional human ATP1F1 active mutant system similar to what they have previously published in EMBO in other tissue settings. This mouse model is very interesting since it provides a genetic approach for the tissue specific inhibition of mitochondrial ATP production that helps to untangle the complicated inter-organ metabolic relationships that occur during the onset of insulin resistance and obesity. Overall this work is very interesting and is extensively characterized at the molecular level using multiple in vitro, in vivo, and ex vivo approaches. They also identify a drug, edaravone, an FDA approved drug in ALS that is thought to be an antioxidant modifier of ROS, which is capable of preventing the onset of some aspects of dysfunction demonstrated in their mouse model. This is intriguing data and adds to the literature on this drug as a potential metabolic therapy that has already been under investigation as a possible modifier of diabetic nephropathy and ischemia reperfusion injury. This manuscript also adds interesting data to the literature on the role of BCAA and lipid metabolism in insulin resistance and obesity.

We thank the Reviewer for appreciating the interest of our contribution.

Major Comments

Q1. There is no assessment of the motor function, are there functional deficits in locomotion, running, strength? It would be interesting to run these mice through some aspects of physiological motor behavior in the absence of HFD treatment to see if at baseline these mice have any physiological impairment of skeletal muscle. Possible examples include grip strength, metabolic cage/activity tracking, open field behavior, running wheel spontaneous exercise measurements, or treadmill forced maximal running testing.

A1. According to Reviewer's request, we have now performed motor function and activity tests to unveil possible functional Skm alterations due to the inhibition of the Skm OXPHOS. No significant alterations in Rotarod test (new Fig 3G) and Open Field test (new Fig 3H and EV4) were noticed between wt and LowOXPHOS animals; however, upon ATP synthase inhibition mice displayed reduced performances in the Grip Force test after fatigue (Fig 3I), suggesting slight muscular weakness.

Q2. I am not convinced by the argument that BCAAs are the main acetyl coA source for lipid droplet formation by the in vitro C2C12 experiments by comparison to pyruvate or citrate sources. It seems that this claim is primarily based on a western blot showing increased phosphorylation of PDH, however this does not demonstrate a complete block of the pathway, and the lack of a difference in lipid droplet formation upon growth in an AA-free medium. However limiting all amino-acids will suppress anabolic metabolism. This experiment is not isolating the effects of BCAA metabolism specifically. It may be interesting to repeat the carbon labeling experiments with labeled glucose or glutamine which may also demonstrate significantly more enrichment of C14 lipids in the setting of ATP1F1 overexpression in C2C12 cells.

A2. We thank the Reviewer for this interesting suggestion. Following her/his indication in the revised version of the manuscript we have better addressed the implication of the increased BCCA catabolism as major source for acetyl-CoA and lipogenesis.

According to reviewer's request, we first investigated glucose metabolism. As previously reported (Formentini et al., 2012), the inhibition of mitochondrial ATP production caused a rewiring of energy metabolism through an increased aerobic glycolysis (new Fig 2J and EV3B), with the aim to maintain Skm ATP levels (new Fig EV3C). This is also accompanied by a slight increase in 14C(u)-glucose uptake (new Fig 2K). However, ATPIF1_{H49K} expressing myocytes showed a reduced total oxidation of 14C(U)-glucose to CO₂ (new Fig 2L), possibly as a consequence of PDH phosphorylation (new Fig 2M) and the redirection of pyruvate to lactate production (Fig 2N). In line with this, the 14C-atoms derived from labeled glucose accumulated in lipids in a similar manner in control and ATPIF1_{H49K} expressing myocytes (new Fig 3E). Therefore, we ruled out glycolysis as the main source of acetyl-CoA during OXPHOS inhibition. However, we support that the increased glycolytic flux prompts the production of glycerol as a backbone for glycerolipid synthesis and thus participates to the lipogenic shift observed.

Interestingly, plasma levels of BCAA were augmented in Low_{OXPHOS} mice compared to wt (new Fig 3A), but no differences in the Skm BCAA amounts were detected between the two genotypes (new Fig 3C), suggesting an increase in BCAA muscular catabolism in Low_{OXPHOS} mice. Consistently, the uptake of 14C(u)-leucine and 14(u)-isoleucine increased in myocytes expressing ATPIF1_{H49K} (new Fig 3B) and the 14C(u)-leucine and 14(u)-isoleucine oxidation to CO₂ was significantly higher upon OXPHOS inhibition (see new Fig 3D). In line with our hypothesis, when the lipid fraction was extracted, a higher concentration of C14-lipids was observed upon OXPHOS inhibition (new Fig 3E).

As further demonstration, we overexpressed the human protein ATPIF1_{H49K} in mouse myocytes in media with or without leucine, isoleucine and valine. In complete media, the inhibition of the ATP synthase during palmitate supplementation triggered a higher number of BODIPY-positive lipid droplets. However, upon BCAA deprivation no differences were observed between the two genotypes (see new Fig 2T and EV3E), highlighting a direct link between BCAA and lipogenesis in our model. The fact that the pharmacological inhibition (5 μ M oligomycin) of the ATP synthase reproduced similar results (new Fig 2U and EV3E, F) suggests that the increase in lipid droplet is a general trait of inhibiting the ATP synthase activity.

We thank the Reviewer for the significant improvement that her/his suggestions have provided to the manuscript. We have now accommodated all these changes in Figure, Result and Discussion sections. (See also the answer to R1, major point 1.)

Q3. Is it possible that the effect of edaravone is simply delaying the onset of insulin resistance in these models? Looking at the curves of the ITT and GTT, it looks like both the WT and the transgenic mice are fairly insulin sensitive upon edaravone treatment at 60 days. Have you measured food intake and activity to see if edaravone is changing either of these parameters to impact the onset of HFD-induced disease in both groups?

A3. We thank the Reviewer for this observation. Following her/his indication, we have repeated the *in vivo* experiment with the four experimental groups (wt, Low_{OXPHOS}, wt+edaravone and Low_{OXPHOS} +edaravone mice) until the setting of IR in all animals. At day 60 of HFD, when nontreated Low_{OXPHOS} mice were already prediabetic, edaravone-treated Low_{OXPHOS} mice showed similar insulin and glucose sensitivity to control (insulin sensitive, see new Fig 7J). New results (new Fig 7J-L) stressed that edaravone delayed the onset of T2D in both genotypes: interestingly, at day 80 of HFD, when both nontreated wt and Low_{OXPHOS} mice were insulin resistant (new Fig 7K), GTT values indicated that edaravone-treated animals were still insulin sensitives (new Fig 7K) and developed T2D only at day 90-100 of HFD (new Fig 7K, L). This is not due to altered food intake caused by edaravone treatment (EV7D) and we are more tempted to ascribe this edaravone property to its ROS-mediated effect on lipogenesis.

Q4. These data suggest that inhibition of mitochondrial ATP production in skeletal muscle is sufficient to increase susceptibility to whole body lipid metabolic changes associated with insulin resistance. It does not, however tackle the necessity of this change, it may be that there are multiple metabolic tissues for which a similar change will result in a similar metabolic phenotype.

A4. We agree with the Reviewer comment. Following her/his suggestion (see also the answers to R1, minor points 6 and 7), we have removed statements about a direct Skm-WAT crosstalk and included a sentence in the Result section about the possibility that the observed body lipid perturbations may be due to whole-body metabolic alterations.

Minor points:

mQ1. In the abstract, branched-chain amino acids is abbreviated as BCCA instead of BCAA.

mA1. We have now corrected this typing error.

mQ2. Transition to lipodystrophy in the second paragraph of the introduction - not sure this is the right term, typically lipodystrophy is reserved for the rare genetic lipodystrophies and isn't really applied to obesity. Why not stay with obesity, or metabolic syndrome? Similarly in the first sentence on page 9 -"ATP-synthase dependent dyslipidemia in muscle" I think a different wording is required since dyslipidemia implies the blood lipid profile.

mA2. We have now changed dyslipidemia to "obesity" or "metabolic syndrome".

mQ3. In the materials section, the statement Mitoq was a "gentle concession" should probably be amended to a gift.

mA3. We have now amended “was a gentle concession” to “was kindly gifted”

mQ4. Figure 1 A there is an n= that doesn't appear to have anything following it.

mA4. We have corrected this typing error.

mQ5. Figure 1 D: is there an increase in liver and WAT mouse endogenous ATP1F by densitometry analysis? If so why do you think that would be?

mA5. No significant changes in densitometry analysis were detected. We have now changed the blot to avoid misunderstanding.

mQ6. Is the first section of Figure 6 E real data or also a schematic like in A? If it is data, are there time points that could be highlighted on the graph to make that distinction more pronounced? Also this data is shown in the bar graph so I am not sure it is necessary.

mA6. We apologize for the misunderstanding. Fig 6E is a representative Seahorse profile for myocytes treated or not with edaravone. New figure now includes standard errors and time points for the message to be clear and accurate.

Referee #3:

The manuscript entitled "Dysfunctional muscle oxidative phosphorylation shunts branched-chain amino acid catabolism into lipogenesis" by Sánchez-González, et al. provides a deep exploration of metabolic consequences of ATP synthase inhibition. The authors generate an interesting mouse model expressing an active form of ATPIF1 in striatal muscle. The mice present with increased weight gain, including increased levels of visceral fat, and develop insulin resistance faster. Muscle fatty acid oxidation decreases while BCAA catabolism increases leading to acetyl-CoA accumulation and reduced OXPHOS. Complementary alterations are observed in adipose tissue and supported by proteomics and metabolic data, including lipidomic analysis. Finally, a screen identifies edaravone as a driver of increased FA oxidation and OCR which prevents diet-induced obesity.

The study is fairly detailed and well-executed. At times the paper reads long, but the authors are covering a lot of metabolic ground. Generally, the respiration and mitochondrial studies are stronger than the lipogenesis data, but the study highlights an interesting contrast between muscle and adipose BCAA catabolism. I have some conceptual questions as well as some technical issues that should be addressed.

We thank the Reviewer for supporting our work.

Q1. Why does increased mitochondrial AcCoA (which is generated by BCAA catabolism) increase tissue AcCoA? For that matter, why does AcCoA increase lipogenesis. FASN and other enzymes are increased, but is this due to acetylation? Inflammation? Please clarify in the manuscript.

A1. We thank the Reviewer for the comment and improvement of the manuscript. Mitochondrial Acetyl-CoA generated by increased BCAA catabolism can reach the cytosol through the citrate/malate/pyruvate shuttle and acts either as direct substrate of the *de novo* lipid synthesis, or contributes to the post-translational modifications (acetylations) that are described to activate the proteins of the pathway. In this regard, following the Reviewer suggestion, we investigated the acetylation state of proteins from the *de novo* lipogenesis pathway. Intriguingly, in Low_{OXPHOS} mice, ACLY resulted highly acetylated, what has been related to the stabilization and activation of the protein, promoting lipid biosynthesis ((Lin et al., Mol Cell, 2013), new Fig 2G). Also inflammation could participate in these events. Consistent with previous data in human myocytes derived from obese subjects (Formentini et al., 2017a), Skm TNF α levels were 270% upregulated and the pattern of cytokines altered when ATP synthase was inhibited (new Fig 3N), what may contribute to inflammation associated with obesity (Ciaraldi et al., 2016). New results also showed a significant increase in iNOS levels in skeletal muscle upon OXPHOS inhibition (new Fig EV3G). Interestingly, quenching mitochondrial ROS with edaravone significantly reduces inflammation and lipogenesis (New Fig 7). We added sentences in this regard in Result and Discussion sections to clarify the message.

Q2. Quantifying BCAA uptake in the ATP1F1H49K cells would be very informative. Is BCAA uptake higher in ATP1F1H49K cells? How does FASN inhibition or culture in delipidated media affect BCAA catabolism?

A2. We thank the Reviewer for this comment. Interestingly, the uptake of 14C(u)-leucine and 14(u)-isoleucine increased in myocytes expressing ATP1F1_{H49K} (new Fig 3B). Moreover, the plasma levels of BCAA were augmented in Low_{OXPHOS} mice compared to wt (new Fig 3A), but no differences in the Skm BCAA amounts were detected between the two genotypes (new Fig 3C), suggesting an increase in BCAA muscular catabolism in Low_{OXPHOS} mice. Consistently, the 14C(u)-leucine and 14(u)-isoleucine oxidation to CO₂ was significantly higher upon OXPHOS inhibition (see new Fig 3D). Interestingly, the treatment with edaravone, that strongly reduced FASN expression (Fig 7D) only slightly decreased 14C(u)-leucine uptake (Fig EV7C).

Q3. The authors attempt to show that BCAAs are essential for LD formation by culturing cells in AA-free media. Is this simply mTOR inhibition/autophagy-driven rather than BCAA-specific?

A3. As the reviewer pointed out, and following the three reviewers' comments (see also answer to R1, major point 1; and R2, major point 2), the revised version of the manuscript includes stronger demonstrations that BCAAs are essential for increased lipogenesis and LD formation.

In complete media, the inhibition of the ATP synthase during palmitate supplementation triggered a higher number of BODIPY-positive lipid droplets. However, upon BCAA deprivation no differences were observed between the two genotypes (see new Fig 2T and EV3E), highlighting a direct link between BCAA and lipogenesis in our model. The fact that the pharmacological inhibition (5 μ M oligomycin) of the ATP synthase reproduced similar results (new Fig 2U and EV3E, F) suggests that the increase in lipid droplets is a general trait of inhibiting the ATP synthase activity. In line with our hypothesis, when myocytes were treated with 14C(u)-leucine and 14(u)-isoleucine and the lipid fraction extracted, a higher concentration of C14-lipids was observed upon OXPHOS inhibition (new Fig 3E). Interestingly, nor the modulation of mTOR or autophagy pathways seemed to participate in these events (Fig EV3G).

Q4. Visceral AT is lower with edaravone treatment. Is muscle whitening decreased?

A4. Yes. No significant changes in Skm whitening (new Fig EV7A) were detected upon edaravone treatment between wt and Low_{OXPHOS} mice. The revised version of the manuscript includes this data in the Result section.

Does this treatment decrease Skm BCAA abundance?

Following the Reviewer's indication, in the revised version of the manuscript we have measured the BCAA abundance in plasma and skeletal muscle from wt and Low_{OXPHOS} mice treated or not

with edaravone(new Fig 3A, 3C and EV7B). A slight reduction in plasma BCAA levels in Low_{OXPHOS} mice was observed after 1-month treatment of edaravone (new Fig EV7B).

Minor:

mQ1. Fig 7C should be PGC1 α .

mA1. We have corrected this typing error.

mQ2. Fig 6E - error bars should be present?

mA2. New figure now includes standard errors and time points for the message to be clear and accurate.

mQ3. Figure layout (letter panels) is tough to follow and should be organized better, though I understand there is a lot of data.

mA3. We apologize if due to manuscript complexity, figures were sometimes disorganized. We have now re-arranged the figures to make the order more logical.

mQ4. Figure 6C poorly depicts the tracing study. Why 3 circles if only 2 isotopes present per molecule?

mA4. We apologize for the inconvenient. We have now corrected the figure to better depict the tracing study.

Thank you for submitting a revised version of your manuscript. It has now been seen by the original referees whose comments are shown below.

As you will see, referee #2 and #3 find that all criticisms have been sufficiently addressed and recommend the manuscript for publication. However, referee #1 still has few concerns that I would like you to address by either including new experimental data (in the case you already have them) or by toning down/expanding the discussion on the specific issues.

In addition to solve the remaining points from referee #1, there are few editorial issues concerning text and figures that I need you to address before we can officially accept the manuscript.

Referee #1:

The revised manuscript by Sánchez-González et al. is greatly improved with regard to experimental evidence as well as presentation and discussion of results. In particular, the authors now provide more direct evidence that increased catabolism of BCAA but not of glucose contributes to the observed lipogenesis phenotypes in low oxphos Skm. Below are a few points that should be addressed.

In the absence of any functional validation, the observed changes in BCAA catabolic enzyme levels remain correlative, which limits the interpretability of the proteomics data. In light of the additional functional experiments, this is not a major concern, but should be discussed.

The authors demonstrate that removal of BCAA from media prevents lipid droplet formation in low oxphos cells (Figure 2T). To support the hypothesis that branched-chain amino acid catabolism is shunted into lipogenesis, the reverse experiment would be more conclusive, i.e. supplementation of BCAA in the absence of other biosynthetic substrates (AA and/or FA).

The authors provide evidence that mTOR signaling and autophagy are not changed in low oxphos Skm (Figure EV3G, H). To conclude that 'modulation of the mTOR or autophagy pathways' does not participate in increased lipogenesis, a direct modulation of these processes would be required.

Referee #2:

I am satisfied

Referee #3:

The authors have addressed all our points and improved the manuscript.

Referee #1:

The revised manuscript by Sánchez-González et al. is greatly improved with regard to experimental evidence as well as presentation and discussion of results. In particular, the authors now provide more direct evidence that increased catabolism of BCAA but not of glucose contributes to the observed lipogenesis phenotypes in low oxphos Skm. Below are a few points that should be addressed.

We thank the Reviewer for her/his positive feedback and for appreciating the improvement of the manuscript.

Q1. In the absence of any functional validation, the observed changes in BCAA catabolic enzyme levels remain correlative, which limits the interpretability of the proteomics data. In light of the additional functional experiments, this is not a major concern, but should be discussed.

The authors demonstrate that removal of BCAA from media prevents lipid droplet formation in low oxphos cells (Figure 2T). To support the hypothesis that branched-chain amino acid catabolism is shunted into lipogenesis, the reverse experiment would be more conclusive, i.e. supplementation of BCAA in the absence of other biosynthetic substrates (AA and/or FA).

A1. We thank the Reviewer for the suggestion. However, due to the COVID-19 situation in Madrid, our University shut down the activity and starting new experiments is not allowed. Therefore, we have now rearranged and toned down the Result and Discussion Sections, highlighting that in the absence of a stronger functional validation, our data point to the participation of the BCAA catabolism in the observed lipogenesis. As an example, in the Discussion Section, we have re-written the sentence *“Because ATP-synthase-mediated LD formation occurred in a BCAA-dependent manner, we support the idea that upon OXPHOS inhibition, muscle cells rely on the oxidation of BCAAs as a major route of acetyl-coA synthesis”*. The new sentence now reads: *“Interestingly, the ATP-synthase-mediated LD formation in myocytes is prevented by removing BCAAs from culture media, what might suggest that upon OXPHOS inhibition, BCAA oxidation participates in acetyl-coA synthesis and increased lipogenesis.”* (Discussion Section, Pag 20).

Q2. The authors provide evidence that mTOR signaling and autophagy are not changed in low oxphos Skm (Figure EV3G, H). To conclude that 'modulation of the mTOR or autophagy pathways' does not participate in increased lipogenesis, a direct modulation of these processes would be required.

A2. We have now changed the sentence *“Interestingly, nor the modulation of mTOR or autophagy pathways seemed to be involved in these events (Fig EV3G, H)”* to: *“Interestingly, we did not find changes in the expression of proteins from mTOR or autophagy pathways in LowOXPHOS mice (Fig EV3G, H)”*. Moreover, we have now added a sentence about the need of further studies on a direct modulation of mTOR or autophagy pathways to exclude their participation in observed lipogenesis (Result Section, Pag. 10).

Referee #2:

I am satisfied

We thank the Reviewer.

Referee #3:

The authors have addressed all our points and improved the manuscript.

We thank the Reviewer.

I am pleased to inform you that your manuscript has been accepted for publication in the EMBO Journal.